# Trapped by simplicity: When Transformers fail to learn from noisy features

**Evan Peters**[1,2]***& Ando Deng**[1] **& Matheus H. Zambianco**[1,2]
**Devin Blankespoor**[1] **& Achim Kempf**[1,2]
[1]University of Waterloo
[2]Perimeter Institute for Theoretical Physics

## Abstract

Noise is ubiquitous in data used to train large language models, but it is not well understood whether these models are able to correctly generalize to inputs generated without noise. Here, we study noise-robust learning: are transformers trained on data with noisy features able to find a target function that correctly predicts labels for noiseless features? We show that transformers succeed at noise-robust learning for a selection of $k$-sparse parity and majority functions, compared to LSTMs which fail at this task for even modest feature noise. However, we find that transformers typically fail at noise-robust learning of random $k$-juntas, especially when the boolean sensitivity of the optimal solution is smaller than that of the target function. We argue that this failure is due to a combination of two factors: transformers' bias toward simpler functions, combined with an observation that the optimal function for noise-robust learning typically has lower sensitivity than the target function for random boolean functions. We test this hypothesis by exploiting transformers' simplicity bias to trap them in an incorrect solution, but show that transformers can escape this trap by training with an additional loss term penalizing high-sensitivity solutions. Overall, we find that transformers are particularly ineffective for learning boolean functions in the presence of feature noise.

## 1 Introduction

Large language models (LLMs) are powerful tools for natural language processing, code generation, scientific research, and reasoning across a wide range of domains. The training data for these models contain noise in the form of stochasticity and different modalities of errors, and yet LLMs trained on these noisy data are often applied in settings where next-token prediction is highly sensitive to noise in the preceding tokens, such as solving arithmetic problems. This raises the question, are transformers actually capable of *noise-robust learning*, i.e. can transformers trained on data with feature noise learn a target function that makes accurate predictions for noiseless data?

Boolean functions of binary input data provide a simplified setting for studying noise-robust learning. Recent results demonstrate that transformers prefer to learn simple boolean functions for next-bit prediction tasks on binary input data (Bhattamishra et al., 2023b; Hahn & Rofin, 2024). An immediate consequence is that the functions learned by these models are robust to input perturbations (i.e. noise) *at evaluation time* (Vasudeva et al., 2025). Yet, less is known about the setting where training data themselves contain feature noise (for example, bitflips randomly applied to the input bitstring). From another angle, Deletang et al. (2024) analyzed natural language modeling in terms of compression in the presence of an inherent amount of randomness found in natural language (Shannon, 1951). However, such analyses neglect the effect of *noise* on information transmission (Shannon, 1948). It is well-known that feature noise induces simpler solutions in ordinary least squares via *attenuation* Fuller (2009), but there is little empirical evidence describing how feature noise affects the learning behavior of transformers in discrete domains, like learning boolean functions. Our goal is to study whether these models are capable noise-robust learning – learning an underlying boolean function from training data with feature noise.

---

*e6peters@uwaterloo.ca

Our main finding is that transformers fail at noise-robust learning for a large class of boolean functions. We attribute this outcome to a combination of two factors: (i) the *simplicity bias* of transformers makes them prefer simple functions that achieve low loss on the training data and (ii) empirical and theoretical arguments that the optimal solution for noise-robust learning is simpler than the target function used to generate the data. Meanwhile, we find that long short-term memory networks (LSTMs), which exhibit less bias towards simple functions, also fail at noise-robust learning, albeit for different reasons. Thus, while a neural network with an inductive bias towards *complex* functions might be capable of noise-robust learning in principle, we show that both transformers (with their simplicity bias) and LSTMs (without a simplicity bias) are poor candidates for this task.

These results imply that transformers are ill-suited to classification and generative modeling in binary domains where feature noise is prevalent, such as learning to decode classical (Kim et al., 2018; Nachmani & Wolf, 2019; Choukroun & Wolf, 2022; Cammerer et al., 2022) and quantum error correcting codes (Torlai & Melko, 2017; Lange et al., 2025; Bausch et al., 2024; Peters, 2025). More broadly, our findings suggest that the simplicity bias of LLMs hurts their ability to learn complex relationships via natural language processing with sufficient feature noise. For instance, models trained on noisy data (e.g. high stochasticity, incorrect grammar or semantics) will likely struggle to perform next-bit prediction for tasks such as arithmetic and discrete mathematics, where each next token depends on preceding noiseless input text according to some sensitive function. This complements observations that feature noise at evaluation time can reduce transformers' ability to do arithmetic and other discrete mathematics (Shi et al., 2023; Abedin et al., 2025). Our work highlights a potential need to mitigate the simplicity biases of large language models, if we hope for these models to learn algorithmic tasks from noisy training data.

**Our contributions** we show mixed results for transformers' performance at noise-robust learning. (i) We find that transformers succeed at this task for sparse parity and majority functions at high rates of feature noise, while LSTMs generally fail at this task even for low levels of feature noise. (ii) We show that transformers fail at this task for random $k$-juntas while simultaneously reaching near-optimal accuracy on noisy validation data. (iii) We propose an explanation for this behavior: We observe that the sensitivity of the optimal solution for noise-robust learning is rarely greater than the sensitivity of the target function, and therefore transformers' simplicity bias will result in a solution that is suboptimal for noiseless evaluation. (iv) We explore this hypothesis by showing that transformers can be trapped by an incorrect solution that achieves similar accuracy as the target function on noisy validation data, and that transformers with a penalty for high-sensitivity solutions can escape this trap.

## 1.1 PRIOR WORK

**Learning boolean functions with transformers:** In an effort to understand the success of contemporary LLMs, significant attention has been given to the ability of transformers to model formal languages (Bhattamishra et al., 2020; Chiang & Cholak, 2022; Strobl et al., 2024), with some results showing shortcomings for modeling certain functions such as PARITY (Hahn, 2020; Bhattamishra et al., 2020). In turn, some work has shown that transformers are biased towards learning low-sensitivity Boolean functions, and that they are robust to label noise (Bhattamishra et al., 2023b; Jonasson et al., 2023; Bhattamishra et al., 2023a). Our work is distinct because we consider learning from examples with feature noise rather than label noise, which has a qualitatively different effect on the learning behavior of language models. The *k-sparse parity* problem is often used to evaluate the learning abilities of transformers (Barak et al., 2023; Michaud et al., 2024), but we show how performance at learning this function is somewhat deceptive in the setting of noise-robust learning.

**Simplicity bias in transformers:** A growing body of evidence shows that neural networks exhibit a bias towards learning simple functions for a variety of domains and simplicity measures (Arpit et al., 2017; Valle-Perez et al., 2019; Kalimeris et al., 2019; Mingard et al., 2020; Cao et al., 2020; Yang & Salman, 2020; Rahaman et al., 2019). Bhattamishra et al. (2023b) showed empirically that transformers tasked with learning boolean functions are biased towards low-sensitivity solutions, compared to other recurrent models such as LSTMs. Later works provided additional theoretical and empirical evidence for this effect (Hahn & Rofin, 2024; Vasudeva et al., 2025). Our work extends and applies these insights in two ways: We demonstrate that LSTMs generally fail to learn boolean functions given noisy input data, while transformers exhibit function-dependent learning abilities. Indeed, we argue that due to their low sensitivity bias, transformers are fundamentally less

capable of learning boolean functions in the presence of feature noise, while such functions might in principle be learnable by a model with high sensitivity-bias.

**Information theory and noisy features:** Information theory provides a natural framework for describing any learner's ability to to predict subsequent or missing tokens of text in terms of the redundancy (or compressibility) of a source of randomness (Shannon, 1951). Accordingly, prior work has related language modeling to compression in the context of Shannon source coding (Teahan & Harper, 2003; Deletang et al., 2024) or Kolmogorov complexity (Sutskever, 2023). However, our analysis differs by considering noisy features produced by some stochastic map acting on noiseless bitstrings, so that our setting is more related to noisy channel coding (Shannon, 1948; 1949) than compression.

## 2 BACKGROUND

We first introduce some notation for dealing with distributions of random variables, and bitstring-valued variables in particular. We usually consider a random length-$n$ bitstring $X := (X_1, \ldots, X_n)$ taking value $x := (x_1, \ldots, x_n) \in \{0, 1\}^n$ uniformly at random. The expected value of a function $f$ with domain $\{0, 1\}^n$ is written $\mathbb{E}_{p_X}[f(x)] := \sum_{x \in \mathcal{X}} p_X(x) f(x)$, or just $\mathbb{E}_x[f(x)]$ when there is no ambiguity. We define the conditional distribution $p_{Y|X}$ of the $n + 1$ bit $Y := f(X)$ generated by applying a boolean function to the *noiseless* input bitstring $X$. Optimal performance at (noiseless) next-bit prediction is related to the next-bit *conditional entropy* of $Y$ given $X$: The entropy $H(X) := \mathbb{E}_x[-\log(p_X(x))]$, loosely, measures the uncertainty in predicting the value of $X$. The conditional entropy $\mathrm{H}(X|Y) := \mathbb{E}_{x,y}[-\log(p_{X|Y}(x|y))]$ describes the uncertainty about the value of $X$ given that $Y$ is known.

We model feature noise using independent, symmetric bitflip errors on uniformly random input bitstrings. This error model is used broadly in both boolean analysis (O'Donnell, 2014), and communication theory (Cover, 1999). We describe bitflips using a random variable $E \in \{0, 1\}^n$, where $\Pr(E_i = 1) := p$. Then, the *noisy bitstring* $Z = X \oplus E$ is generated by adding $E$ to the *noiseless* bitstring $X$ (mod 2), and induces a distribution $p_Z$. We generate a training data point $(Z, Y)$ for next-bit prediction with noisy features as follows: (i) Sample a noiseless bitstring $X$ according to $p_X$, (ii) generate the next bit $Y = f(X)$, (iii) apply iid bitflip errors $E^n$ to create noisy features $Z$, and the goal of the learner is to predict $Y$ given $Z$ with $(Z, Y) \sim p_{ZY}$. Noisy validation data are generated in the same way, while noiseless test data $(X, Y)$ are sampled directly from $p_{XY}$. We define the *noisy generalization error* of a boolean function $g$ with respect to a label function $f$ (used to generate labels $Y = f(X)$) as

$$\mathrm{err}_f(g) := \Pr_{X,Z}(g(Z) \neq f(X)) = \Pr_{Z,Y}(g(Z) \neq Y), \tag{1}$$

where $\Pr_X(\cdot)$ denotes the probability with respect to $X \sim p_X$. We are mainly interested in a model's ability to learn the function $f$ after training only on noisy training data $(Z, Y)$. The *noiseless generalization error* of a model $g$ evaluated on noiseless data is computed as $\Pr_X(g(X) \neq f(X))$. A learning algorithm succeeds at noise-robust learning if it learns a boolean function $g$ with small noiseless generalization error, after being trained to minimize (empirical estimates of) $\mathrm{err}_f(g)$. Throughout this work, we *only* consider training data with noisy features $(Z, Y)$ where $Z$ is a noisy version of $X$ while $Y$ is unaffected by label noise.

We will compare the noise-robust learning capabilities of self-attention network transformers (SANs) (Vaswani et al., 2017) to LSTMs, continuing a recent line of investigation into the relative advantages and behaviors of these models (Bhattamishra et al., 2023b). Many of our experiments involve the parity function, $\mathrm{PARITY}(x) := \mathrm{wt}(x) \mod 2$, and the majority function $\mathrm{MAJ}(x)$ which outputs 1 if and only if $\mathrm{wt}(x) \geq n/2$, where $\mathrm{wt}(x)$ denotes the Hamming weight. Importantly, $\mathrm{MAJ}$ is an imbalanced function when $n$ is even. When relevant, we will use a subscript to refer to the input length (e.g. $\mathrm{MAJ}_n$). We will often consider *sparse* versions of these functions, whose output only depends on a subset of $k \leq n$ input bits. We will denote this by an $(n, k)$ pair, e.g. $\mathrm{MAJ}(n, k)$ computes the majority for a specific subset of $k$ bits.

Performing experiments using boolean functions dependent on relatively few bits allows us to compare models' performance to an optimal prediction rule. We denote a Bayes-optimal predictor for the distribution $p_{ZY}$ as $f_N^* : \{0, 1\}^n \to \{0, 1\}$, to be a boolean function that satisfies

$$\mathrm{err}_f(f_N^*) \leq \mathrm{err}_f(g) \tag{2}$$

for all $g : \{0, 1\}^n \to \{0, 1\}$. The choice of optimal predictor is not unique in general (for instance, when $f(x) = x_1$, $f_N^*(x)$ will not be influenced by $x_2 \ldots x_n$). We choose to evaluate a particular optimal predictor with the following formula:

$$f_N^*(x) := \text{sign}(T_{1-2p} f(x)), \tag{3}$$

where $T_\rho g(x) := \mathbb{E}_{Z|X=x}[g(Z)]$ denotes the *noise operator* with respect to the distribution $p_{XZ}$ with bitwise correlation $\rho := \mathbb{E}[x_i z_i]$ O'Donnell (2014). We define $\mathcal{F}_N^*(f)$ to be the set of all boolean functions $h$ such that $\text{err}(h) = \text{err}(f_N^*)$. We defer further background on Boolean functions to Appendix A. As mentioned before, the task of predicting $f(x)$ given $z$ is rooted in noisy channel coding. Specifically, the best possible accuracy $\text{err}(f_N^*)$ is bounded from above and below by (monotone decreasing) functions of the *next bit (conditional) entropy* (Feder & Merhav, 1994):

$$\Phi^{-1}\left(\text{H}(Y|Z)\right) \le \text{err}_f(f_N^*) \le \phi^{-1}\left(\text{H}(Y|Z)\right). \tag{4}$$

Here, $\phi$ is an invertible piece-wise linear function, while $\text{H}(Y|Z) \le \Phi(\text{err}_f(f_N^*))$ is Fano's inequality (see Appendix A.2). The inequalities in Eq. 4 are analogous to upper (Candes & Tao, 2006; Donoho, 2006) and lower bounds (Peters, 2024) for certain learning tasks on continuous domains.

The dependence of next-bit prediction accuracy with noisy features on the next-bit conditional entropy $\text{H}(Y|Z)$ captures a broader relationship between next-token prediction and noisy channel coding (Shannon, 1948). We model next-token prediction as a communication process between a sender Alice and a receiver Bob, using a finite alphabet $\Lambda$. Alice first chooses a token of information $Y = f(X) \in \Sigma$ that she wishes to communicate to Bob. She then encodes this token into the space of token sequences $X \in \Lambda^n$. This encoding process contains some randomness (e.g. there are many ways to construct sentences that are semantically equivalent), as well as noise (grammatical or syntactical errors), and so the map $Y \to X$ need not be one-to-one or even deterministic. Bob receives noisy bitstring $Z \in \Lambda^n$ due to this combination of noise and stochasticity, and his goal is to decode the token $f(X)$ of Alice's message given the string of noisy tokens $Z$ *without knowledge of her encoding scheme*. Thus, learning next-token prediction from noisy features is firmly rooted in noisy channel coding, and is distinct from (but complements) alternative models based on source coding (compression) (Deletang et al., 2024) or Kolmogorov complexity (Sutskever, 2023). Since Bob does not a priori know Alice's encoding scheme, he may not be capable of decoding her messages even in the noiseless setting. Eq. 4 therefore relates next-bit entropy to limitations on generalization performance in language modeling, and is related to noisy channel capacity in the asymptotic limit. In contrast, Bob's ability to learn $f$ from noisy features depends on properties of the noise and $f$, which we will return to in Section 3.1.

## 3 TRANSFORMERS SUCCEED AT NOISE-ROBUST LEARNING OF SPARSE PARITIES AND ODD MAJORITIES

We now show that for a certain class of functions, transformers are able to learn a noiseless function $f$ when trained entirely with noisy features. This moves beyond prior work showing that transformers learn functions robust to noise at evaluation time (Vasudeva et al., 2025). Our experiments show that transformers are more robust than LSTMs at learning PARITY$(n, k)$ and MAJ$(n, k)$ (but only when $k$ is odd), when trained on data with iid bitflip noise applied to input features. We tested two kinds of boolean functions: sparse majorities and sparse parities. For each choice of bitflip rate $p$, we generated a size $N$ dataset of $(Z, f(X))$ pairs, and then trained an LSTM or transformer to predict the label for each point. We do not apply label noise to any of our data. Since a model's ability to learn is sensitive to initial parameter choice and hyperparameters, we repeated this process 300 times for each model with random initializations and random hyperparameters. We constructed the set of possible hyperparameters by iterative grid search while training each model on noiseless data, and then expanded this set slightly to account for hyperparameter optimality changing with noise strength. We provide detailed experiment descriptions in Appendix B.

Fig. 1 shows the relative performance of LSTMs versus SANs on learning several commonly-tested boolean functions with feature noise. We considered the sparse majority functions MAJ$(20, 5)$, MAJ$(40, 5)$, MAJ$(50, 3)$ with $N = 2000$ noisy training data. In both cases, LSTMs and SANs reliably learn with zero feature noise. At modest noise strengths, the *best* of 300 LSTMs performs worse than the *median* transformer (which performs close to the information-theoretic optimal). We also compare both models' abilities to learn PARITY$(n, k)$. However, sparse parity is significantly

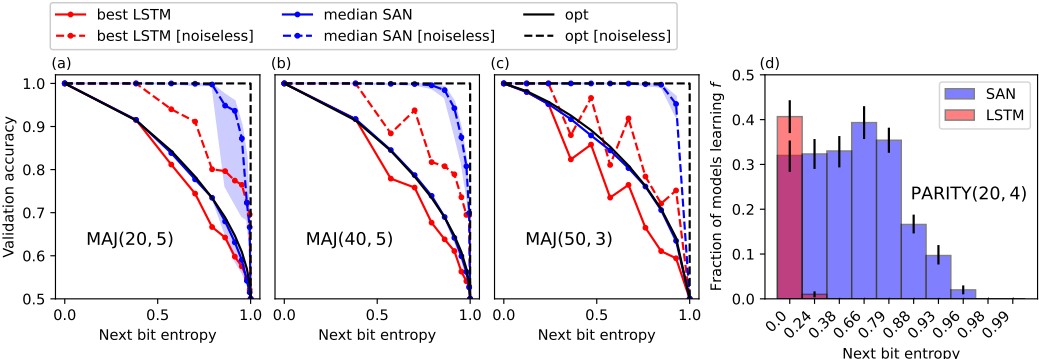

Figure 1: **Transformers learn MAJ$_n$ (with odd $n$) and PARITY robustly from noisy features**. (a-c) For MAJ$(20, 5)$ and MAJ$(40, 5)$, the *median* transformer (SAN) reliably outperforms the *best* LSTM across 300 training runs with a variety of hyperparameters tuned to optimize both architectures' success probability. Validation accuracy approximates $\text{err}_f(\hat{f})$ using 10000 examples, where $\hat{f}$ is either an LSTM or SAN prediction rule. Each point on the solid lines represents the best (median) LSTM (SAN) from 300 training experiments. (d) While both LSTMs and SANs fail in a large fraction of training experiments learning PARITY$(20, 4)$ with feature noise, transformers successfully learn PARITY$(20, 4)$ (defined as achieving noiseless accuracy $\geq 95\%$) more often than LSTMs, even when both architectures perform comparably at zero noise rate. See Appendix B for experiment details.

harder to learn: (i) LSTMs generally fail to learn PARITY$(n, k)$ for $n > 20$ even with noiseless inputs (Bhattamishra et al., 2023b), and (ii) learning parity with feature noise rate $p$ corresponds to an instance of *learning parity with noise* (Blum et al., 2003), an intractable learning problem. For $f := \text{PARITY}(20, 4)$, we compute the fraction of models that learned $f$ from noisy training data since the training process is highly sensitive to initial parameter choices. Among SANs and LSTMs that achieve comparable performance on learning PARITY$(20, 4)$ in the noiseless setting, only SANs are capable of learning from noisy input features. We also found that this behavior extends to sparse multitask parities (Michaud et al., 2024), for which transformers succeeded at noise-robust learning while all LSTMs failed. Thus, for the commonly-studied PARITY and MAJ functions, we find that transformers achieve better learning outcomes across a variety of hyperparameter choices and initializations.

### 3.1 NOISE ROBUSTNESS VERSUS SIMPLICITY BIAS

What explains transformers' impressive robustness to feature noise compared to LSTMs? Fig. 1 demonstrated that transformers can learn from data with feature noise more robustly than LSTMs for a few specific functions. One explanation for this behavior is the observed *simplicity bias* of deep neural networks (Arpit et al., 2017; Valle-Perez et al., 2019; Mingard et al., 2020; Cao et al., 2020), as recent works demonstrate that transformers tend to learn simpler boolean functions compared to LSTMs without any explicit regularization (Bhattamishra et al., 2023b; Hahn & Rofin, 2024). Sensitivity is a common measure of simplicity in the context of boolean functions (Kahn et al., 1988). The *sensitivity* of a boolean function $f$ is defined as

$$\text{I}[f] := \sum_{i=1}^{n} \Pr_{x \sim \{0,1\}^n} (f(x) \neq f(x^{\oplus i})), \tag{5}$$

where $x^{\oplus i}$ denotes $x$ with a bitflip at location $i$. Simplicity bias helps explain why transformers trained on noiseless data enjoy robustness to feature noise at *evaluation time*, since small test error on noisy data combined with transformers' robustness implies small noisy generalization error (Eq. 1): For uniformly distributed $x \in \{0, 1\}^n$ and iid bitflip noise with probability $p$, a function $\hat{f}$ obeys $\Pr_{x,z}(\hat{f}(x) \neq \hat{f}(z)) \leq p\,\text{I}[\hat{f}]$ (O'Donnell, 2014). Ordinarily, a transformer is trained to achieve low error on *noiseless inputs*, such that $\Pr_x(\hat{f}(x) \neq f(x)) = \epsilon$ is small. Then, a simple triangle

inequality gives

$$\text{err}_f(\hat{f}) \leq \epsilon + p\,\text{I}[\hat{f}]. \tag{6}$$

From this, we see that a transformer's simplicity bias (preference for $\hat{f}$ with small $\text{I}[\hat{f}]$) implies small error in predictions on noisy features at evaluation time. This intuitively explains observations that transformers trained on noiseless data can accurately classify noisy examples at evaluation time (Zhou et al., 2022), and has been referred to as *noise robustness* in other literature (Vasudeva et al., 2025).

However, in our setting we are concerned with *training models on noisy features*, which is a qualitatively different than what is typically studied for boolean learning problems. Specifically, we ask: Will transformers trained on noisy features learn to make accurate predictions on noiseless data at evaluation time? The general answer is **no**. Our first clue that transformers can fail at noise-robust learning is that the functions PARITY$(n, k)$ and MAJ$(n, k)$ analyzed above are actually special boolean functions that happen to be optimal for prediction on both noiseless and noisy features. The following proposition summarizes several results of (Weinberger & Shayevitz, 2018) demonstrating how the majority and parity functions functions are qualitatively special among boolean functions:

**Proposition 1.** For each function $f \in \{\text{MAJ}_n, \text{PARITY}\}$ ($n$ odd), $f$ is optimal for prediction on noisy features data, i.e. $f = f_N^*$.

In the terminology of Weinberger & Shayevitz (2018), the majority and parity functions are *self-predicting*, as each function achieves optimal test error when evaluated on its respective noisy input distribution $(Z, f(X))$ and can therefore, in principle, be learned by ordinary loss minimization (the self-predicting property immediately extends to sparse versions of each function). In this way, MAJ$(n, k)$ with odd $n$ and PARITY$(n, k)$ function are uniquely easy to learn with feature noise. We will now argue that for a typical boolean function $f$, it is instead true that $f \neq f_N^*$. Therefore, we should expect that SANs and LSTMs will fail to learn $f$ from noisy inputs, though we show an example of mitigating this shortcoming via a tailored loss function.

## 4    TRAPPING TRANSFORMERS WITH SIMPLICITY BIAS

Section 3.1 suggested that transformers can learn boolean functions robustly in the presence of feature noise for several commonly-considered functions. We now show that this behavior does not hold in general, and should even be considered atypical. We will show that transformers generally fail at noise-robust learning due to a combination of their simplicity bias, and the observation that the optimal prediction rule for noisy data on average has lower sensitivity than the optimal prediction rule for noiseless data for randomly $k$-juntas with small $k$. Thus, whenever a model with low sensitivity bias is trained via risk minimization on noisy examples $(Z, f(X))$ on such functions, the model will typically fail to learn the target function $f$ without further intervention.

Intuitively, feature noise should cause an optimal predictor $f_N^*$ to be simpler than $f$, by some measure of simplicity. For example, if $f$ is an imbalanced function and $p$ is sufficiently large, $f_N^*$ will be a constant function. The relative simplicity of $f_N^*$ compared to $f$ is also observed in least-squares regression as *attenuation*, wherein feature noise decreases the learned slope in ordinary least squares regression (Fuller, 2009). Similarly, feature noise can be understood to act as a regularization term in learning problems (Bishop, 1995; Wager et al., 2013). Whenever $f_N^* \neq f$, a model minimizing validation error by finding some $\hat{f} \in \mathcal{F}_N^*(f)$ will forgo the possibility of learning $f$. In this case, it might still be possible to learn $f$ noise-robustly, but only if the model has inductive biases towards learning $f$ rather than alternative solutions $g$ with $\text{err}_f(g) < \text{err}_f(f)$. Conversely, the simplicity bias of transformers can actually harm their ability to do noise-robust learning since the optimal solution $f_N^*$ for noise-robust learning often has lower sensitivity than the target function $f$, which we make concrete with the following proposition:

**Proposition 2.** Let $f : \{0, 1\}^n \to \{0, 1\}$ be a function sampled uniformly at random from the set of all boolean functions on $n$ bits. Then, for sufficiently large $n$ the optimal predictor $f_N^*(x) := \text{sign}(T_{1-2p}f(x))$ has average sensitivity

$$\mathbb{E}_f[\text{I}[f_N^*]] \approx \frac{n}{\pi} \arccos\left(\frac{2p(1-p)}{p^2 + (1-p)^2}\right), \tag{7}$$

and in particular for $p \in (0, 1/2]$,

$$\mathbb{E}_f[\text{I}[f]] > \mathbb{E}_f[\text{I}[f_N^*]] \tag{8}$$

We prove Proposition 2 in Appendix A.4. Importantly, Eq. 8 states that for uniformly random boolean functions on sufficiently many bits, the sensitivity $f$ is larger than the sensitivity of the optimal predictor $f_N^*$ on average. We observe that this inequality typically holds even for small $n$: Fig. 2 shows that $I[f] \geq I[f_N^*]$ holds for random $k$-juntas ($k \in (5, 6, 7, 8)$), and additional simulations find no violations of the inequality. On the other hand, there *do* exist functions for which $I[f] < I[f_N^*]$, and so $I[f] \geq I[f_N^*]$ can only hold as an average case statement. We discuss these details further in Appendix A.4.

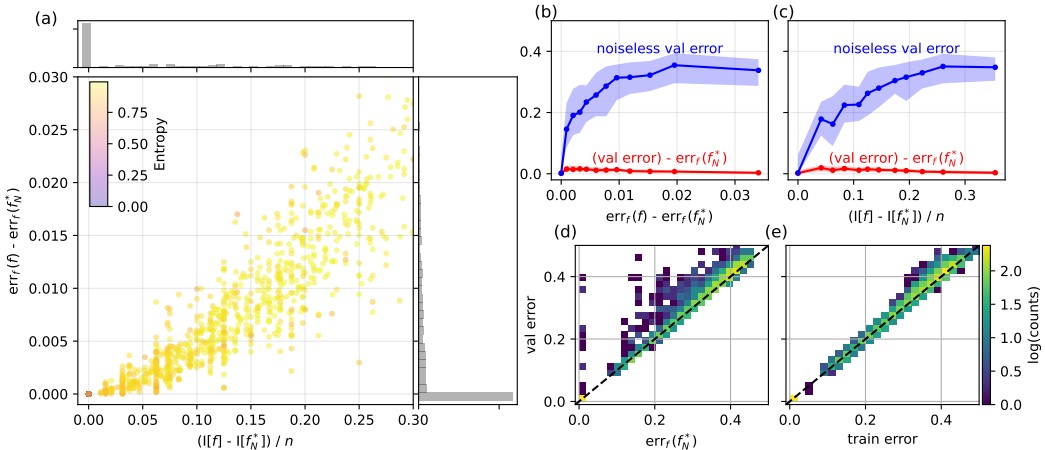

Figure 2: **Transformers generally fail at noise-robust learning for random $k$-juntas, and perform worse as the difference in sensitivity and validation error for $f$ versus $f_N^*$ grows.** (a) Each point represents a randomly sampled $k$-junta $f$ (3200 total). Every randomly sample $f$ obeys $I[f] \geq I[f_N^*]$, while by definition $\mathrm{err}_f(f) \geq \mathrm{err}_f(f_N^*)$. Minimizing validation error in noise-robust learning will only succeed for functions near the bottom of the plot, while a training algorithm with low sensitivity bias will only succeed for points near the left of the plot. By Prop. 1, $\mathrm{MAJ}_n$ (odd $n$) and $\mathrm{PARITY}$ are represented by the coordinate $(0, 0)$. (b-c) Transformers only succeed at noise-robust learning when $I[f] \approx I[f_N^*]$ and $\mathrm{err}_f(f) \approx \mathrm{err}_f(f_N^*)$ (across 3200 learning experiments). Histograms of models' final validation error, train error, and optimal error demonstrate that noise-robust learning fails despite (d) near-optimal performance with (e) little overfitting. See Appendix B for additional experimental details.

Fig. 2 demonstrates that the sensitivity difference between $f$ and $f_N^*$ is usually large, with respect to random boolean functions. Correspondingly, across 3200 randomly-generated $k$-sparse boolean functions ($k$-juntas), we find that transformers tend to perform worse at noise-robust learning as the difference $(I[f] - I[f_N^*])$ grows. This is despite – or because of – achieving near-optimal validation error with minimal overfitting, and demonstrates a correlation between the transformer's worsening performance and decreasing sensitivity of the optimal solution for noisy data. However, this relationship is potentially confounded as the difference $(\mathrm{err}_f(f) - \mathrm{err}_f(f_N^*))$ also grows, which will make the transformer less likely to learn $f$ via loss-minimization regardless of $I[f]$.

To isolate effect of simplicity bias on transformers' and LSTMs' ability to do noise-robust learning, we design a controlled experiment involving a trap function $f$ such that $\mathrm{err}_f(f_N^*) \approx \mathrm{err}_f(f)$, but $I[f_N^*] \ll I[f]$. In this case, a learning algorithm could learn $f$ from noisy data in principle, but will fail if it is biased towards lower sensitivity functions. Indeed, Figs. 3(a-c) demonstrate that transformers fail to learn $f$ and instead converge towards the trap function $f_N^*$. LSTMs also fail to learn $f$, but instead due to overfitting on training data. Thus, SANs and LSTMs both struggle with noise-robust learning, but fail to learn $f$ from noisy data for completely different reasons.

A neural network might be capable of escaping the trap and learning $f$ if we were to replace its preference for simple functions with a preference for *complex* functions. To test this possibility, we ran learning experiments for the same trap function but with an additional term in the loss function that penalizes low-sensitivity solutions. We add an additional loss term approximating $-\lambda I[\hat{f}]$ at each training step, which achieves a similar effect to penalizing high-weight components of the function's Fourier spectrum (Gorji et al., 2023). For a narrow choice of $\lambda$, transformers were capable of es-

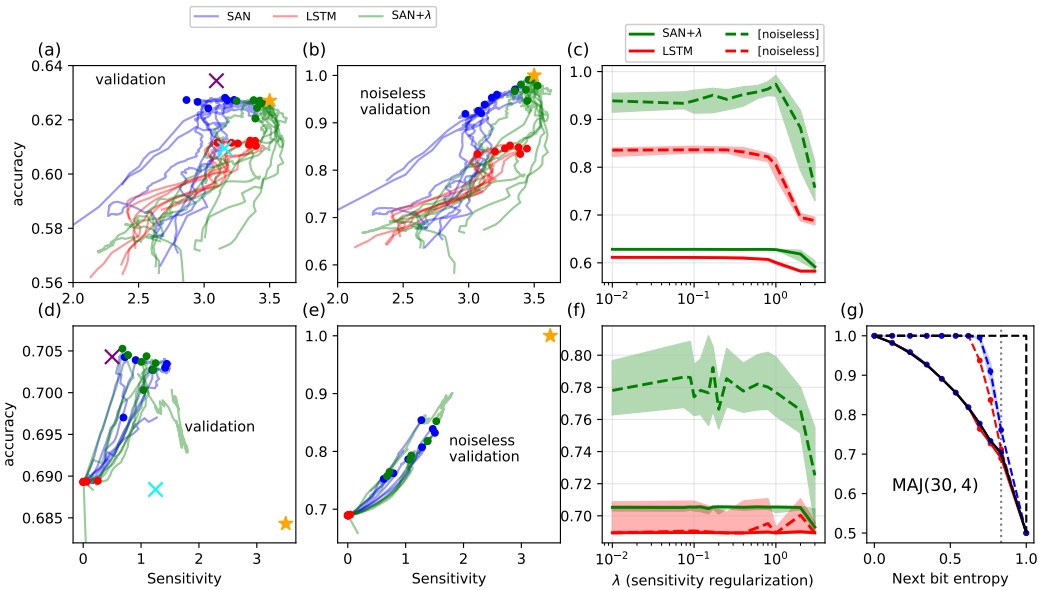

Figure 3: **LSTMs and transformers fail to learn $f$ from training data with feature noise in distinct ways.** (a)-(b) We consider a particular trap function $f$ such that $\text{err}_f(f_N^*) \approx \text{err}_f(f)$, while $\text{I}[f_N^*] \ll \text{I}[f]$. Blue and red lines show (smoothed) training dynamics of transformers and LSTMs trained on noisy inputs across a variety of hyperparameters and initializations. Each point represents a (learned) boolean function. Transformers approach optimal validation accuracy ($\times$) while RNNs perform no better than memorization of training data ($\times$), and both models fail to learn $f$ ($\star$). However, an explicit sensitivity penalty in the loss function $\lambda\,\text{I}[\hat{f}]$ ($\lambda = 1$) allows transformers to learn $f$ (green lines) (c) There is a clear optimum $\lambda$ for learning $f$ from noisy data using sensitivity penalty in the loss. (d-f) This behavior does not extend to functions where $\text{err}(f_N^*) \ll \text{err}_f(f)$, for example MAJ$(30, 4)$ with $p = 0.32$, for which $f_N^*$ is a heavily biased function. (g) Overall, transformers do not outperform LSTMs at learning MAJ$(n, k)$ with even $n$ (shown: MAJ$(30, 4)$) with feature noise. See Appendix B.3.2 for additional details.

caping the learning trap with this complexity bias (Fig. 3c), though this outcome depends on a good choice for $\lambda$, which may not be practical to optimize. Furthermore, a learning algorithm is unlikely to learn $f_N^*$ when $\text{err}_f(f_N^*) \ll \text{err}_f(f)$, even with a strong complexity bias. To demonstrate, we return to $f := \text{MAJ}(n, k)$ ($n$ even), for which $f_N^* \neq f$ in general since $f$ is imbalanced. Fig. 3(d-f) shows that transformers typically fail to learn MAJ$(30, 4)$ from noisy features (for many penalty parameters $\lambda$), while at the same time LSTMs become more capable of learning the lower-sensitivity $f_N^*$. As a result, the performance gap between the median SAN and best LSTM for learning MAJ$_n$ with even $n$ almost completely disappears (Fig. 3e).

## 5 DISCUSSION

**Limitations:** Our noise model and input data are limited in several ways: We have only considered noise in the form of independent bitflip errors on uniformly random input bitstrings, and it is unclear if our findings would extend to other forms of noise. Many of the effects we observed depend on the condition that $f_N^* \neq f$, which in turn requires relatively high noise rates that may not be present in natural datasets. While we have not considered label noise, previous analyses suggest that this would not affect the qualitative behavior of our models (Bhattamishra et al., 2023b).

Here we have studied independent and identically distributed bitflip noise, a toy model for noise occurring in real world data. Our observations should extend to other noise processes: For example, the mechanism by which $T_\rho$ decreases the sensitivity of a boolean function does not depend on identically distributed errors, and so the inequality $\text{I}[\hat{f}_N^*] \leq \text{I}[f]$ should hold with high probability even for non-iid noise models. But here is no reason to expect that this inequality will hold for noise

models with arbitrary correlations between bitflips (see Appendix A.5). Moreover, bitflips lead to a qualitatively different form of noise than randomness introduced by generating bit sequences recurrently according to some stochastic process. Future work will be necessary to determine the extent to which complex and realistic noise models obey $I[f] \geq I[f_N^*]$.

Our experiments with noise-robust learning involved randomly-generated $k$-juntas, which may not be representative of real noise-robust learning problems. Our trapping experiment using sensitivity penalties succeeded under tightly controlled conditions, though this experiment is provided only as a demonstration and this particular strategy may fail in general. It is likely that more sophisticated techniques will be needed to improve transformers' ability to do noise-robust learning when $f_N^*$ is much more accurate than $f$.

**Future work:** Our results suggest that transformers' simplicity bias has practical, negative consequences for learning boolean functions from training data with noisy features. Do these consequences extend to domains with more complex inputs such as natural language processing? In certain natural language tasks like modular arithmetic (Liu et al., 2022), one hopes that transformers can learn input-sensitive discrete functions after training on text that contains both errors and stochasticity (i.e. entropy of written language). Our findings suggest that this may be impossible for sufficiently high-entropy inputs, since optimal prediction on noisy features corresponds to a low complexity predictor on the training data. Our work complements recent observations that LLMs tend to perform poorly at math when exposed to noise *at evaluation time* (Shi et al., 2023; Abedin et al., 2025), and suggests a need for further experiments to examine how noise and stochasticity in training data affect LLMs reasoning abilities at evaluation time. Indeed, the mismatch between $f$ and $f_N^*$ in our experiments suggests that reducing model loss to the next-token conditional entropy may actually be detrimental for learning precise concepts from natural text.

## 5.1 CONCLUSION

We have shown that transformers outperform LSTMs for learning sparse parities and (odd-length) sparse majorities in the presence of feature noise. But we show that transformers fail at noise-robust learning of boolean functions more generally, and use controlled experiments with modified loss functions to connect this failure specifically to transformers' bias towards learning simple boolean functions. Our analysis suggests that transformers may be particularly unsuitable for learning sensitive functions in the presence of feature noise.

## CODE AVAILABILITY

Code to reproduce experiments an analysis is available on Github: `https://github.com/peterse/noise-robust-boolean-learning/`.

## ACKNOWLEDGMENTS

We thank Cyril Goutte and Jackie Lo for helpful discussions. This work was supported by the Applied Quantum Computing Challenge Program at the National Research Council of Canada. Research at Perimeter Institute is supported in part by the Government of Canada through the Department of Innovation, Science and Economic Development and by the Province of Ontario through the Ministry of Colleges and Universities. This work was supported in part by the Dieter Schwarz Foundation.

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

# APPENDICES

## APPENDIX A   BACKGROUND

### A.1   BOOLEAN ANALYSIS

Unless otherwise noted, we will always consider uniform distributions of bitstrings $x \in \{0,1\}^n$, i.e. $p_X(x) = 2^{-n} \, \forall x$. We can represent bitflips using an iid bitflip random variable $E = (E_1, \ldots, E_n)$ where each $E_i$ takes values in $\{0,1\}$ with $\Pr(E_i = 1) = p$. This means that $\Pr(\mathrm{wt}(E) = w) = (1-p)^{n-w} p^w$, for instance. Then, $Z := X \oplus E$ represents the string $X$ that has undergone a bitflip at each location where $E$ is nonzero. The variables $X$ and $Z$ are not independent, but $X$ and $E$ are independent (written $X \perp E$). We can compute relationships of $X$ and $Z$ using this fact, for example:

$$p_{X|Z}(x|z) = \frac{\Pr_{X,Z}(X=x, Z=z)}{p_X(x)} \tag{9}$$

$$= \frac{\Pr_{X,E}(X=x, E=x \oplus z)}{p_X(x)} \tag{10}$$

$$= p_E(x \oplus z) \tag{11}$$

The sensitivity of $f$ at $x$ is defined as the number of bit positions such that a single bitflip of $x$ changes the value of $f$:

$$s(f, x) := \sum_{i=1}^{n} \mathbb{I}\{f(x) \neq f(x \oplus e_i)\}. \tag{12}$$

On the other hand, if we fix a single bit $i$, then the influence of bit $i$ is defined as

$$\mathrm{Inf}_i[f] := \Pr_x(f(x) \neq f(x^{\oplus i})). \tag{13}$$

The average sensitivity (or "total influence") of $f$ is then defined as the average sensitivity over all inputs

$$\mathrm{I}[f] := \mathbb{E}_x[s(f, x)] = \frac{1}{2^n} \sum_{x \in \{0,1\}^n} s(f, x) = \sum_i \mathrm{Inf}_i[f]. \tag{14}$$

This takes values in the range $[n]$, and thus $\mathrm{I}[f]/n$ gives the likelihood over random inputs that a single bitflip changes the output value of $f$.

If we alternatively represent bits $\{0,1\}$ as $\{1,-1\}$, we represent noise by considering pairs $(x,z)$ with $x$ sampled uniformly at random from $\{-1,1\}^n$ and $z$ sampled conditionally such that each bit satisfies $\mathbb{E}[x_i z_i] = \rho$. In this case, we define the noise operator $T_\rho$ according to $T_\rho f(x) := \mathbb{E}_{z \sim_\rho x}[f(z)]$, where $z \sim_\rho x$ denotes sampling $z$ conditionally on $x$ in the way described above.

We will sometimes use Fourier analysis of Boolean. For a subset $S \subseteq [n]$, the Fourier coefficient of $f : \{1, -1\}^n \to \mathbb{R}$ at subset $S$ is defined

$$\hat{f}(S) := \sum_{x \in \{-1,1\}^n} f(x) \chi_S(x) \tag{15}$$

where $\chi_S(x) := \prod_{i \in S} x_i$.

### A.2   INFORMATION THEORY

We briefly introduce concepts from information theory, in order to motivate how next-bit conditional entropy closely describes the hardness of next-bit prediction. Consider a random variable $X$ taking values $x \in \mathcal{X}$ with probability $p_X(x)$. The entropy of $X$ is defined as

$$\mathrm{H}(X) = -\sum_{x \in \mathcal{X}} p_X(x) \log p_X(x). \tag{16}$$

For a joint distribution $p_{X_1 X_2}$ over the pair of random variables $(X_1, X_2)$, the *conditional entropy* of $X_2$ given $X_1$ is

$$\mathrm{H}(X_2|X_1) = \mathrm{H}(X_1 X_2) - \mathrm{H}(X_1) = -\sum_{x_1,x_2} p_{X_1 X_2}(x_1, x_2) \log p_{X_2|X_1}(x_2|x_1). \tag{17}$$

The following theorem from Ref. Feder & Merhav (1994) shows how the error of a Bayes-optimal next-token predictor is tightly controlled by the entropy of the next bit:

**Theorem 3. (Feder, 1994)** Define the piecewise function $\phi_N : [0,1] \to \mathbb{R}$ and $\Phi_N : [0,1] \to \mathbb{R}$ as

$$\phi_N(\lambda) = \left\{ a_k(\lambda - \tfrac{k-1}{k}) + \log(k), \quad \tfrac{k-1}{k} \leq \lambda \leq \tfrac{k}{k+1} \right. \tag{18}$$

$$\Phi_N(\lambda) = h_2(\lambda) + \lambda \log(N-1) \tag{19}$$

with $a_k = k(k+1)\log((k+1)/k)$ for $k = 1 \ldots N$. Let $X$ and $Y$ be random variables be a random variable taking values in alphabets $\mathcal{X}$ and $\mathcal{Y}$ respectively, with $|\mathcal{X}| = N$. Define the error probability of an optimal estimator as $p_e^*(Y|X) := \min_{\hat{Y}:\mathcal{X} \to \mathcal{Y}} \Pr(\hat{Y}(X) \neq Y)$. Then,

$$\Phi_N(p_e^*(Y|X)) \geq H(Y|X) \geq \phi_N(p_e^*(Y|X)) \tag{20}$$

Evidently, the optimal noisy generalization error $\mathrm{err}_f(f_N^*)$ is close to zero if and only if the corresponding entropy $\mathrm{H}(f(X)|Z)$ is close to zero, and $\mathrm{err}_f(f_N^*)$ approaches $1/2$ as $\mathrm{H}(f(X)|Z)$ approaches 1.

### A.3  OPTIMAL NEXT-BIT PREDICTION

In this section, we show that $f_N^*$ from Eq. 1 minimizes $\mathrm{err}_f$, and provide a combinatorial proof for part of Proposition 1 as an alternative to Ref. Weinberger & Shayevitz (2018). We will always assume a uniform distribution of input bitstrings $x \in \{0,1\}^n$, and a conditional distribution for noisy bitstrings $z$ based on i.i.d. bitflips applied to each bit of $x$ with probability $p$. The set of (Bayes-)optimal predictors $\mathcal{F}_N^*(f)$ for a set of boolean-labeled data with corrupted inputs $\{(Z_i, f(X_i))\}$ is given by all $g$ satisfying

$$\Pr_{x,z}(g(z) = f(x)) \geq \Pr_{x,z}(h(z) = f(x)) \tag{21}$$

for all boolean functions $h$. We will now show that $f_N^*$ defined in Eq. 2 of the main text is an optimal predictor:

**Lemma 4.** For any symmetric bitflip rate $p \in [0, 0.5)$, then for $\rho = 1 - 2p$ we have

$$f_N^* := \mathrm{sign}(T_\rho f) \in \mathcal{F}_N^*(f) \tag{22}$$

*Proof.* Eq. 21 is equivalent to

$$g(z) = \begin{cases} 1, & \Pr_x(f(x) = 1|Z = z) \geq 1/2 \\ -1, & \text{else} \end{cases} \tag{23}$$

Compare this to

$$T_\rho f(x) := \mathbb{E}_{z \sim N_\rho(x)}[f(z)] \tag{24}$$

$$= \sum_z p(z|x) f(z) \tag{25}$$

$$= \sum_{z:f(z)=1} p(z|x) - \sum_{z:f(z)=-1} p(z|x) \tag{26}$$

$$= \Pr_z(f(z) = 1|X = x) - \Pr_z(f(z) = -1|X = x) \tag{27}$$

$$= \mathbb{E}_z[f(z)|X = x] \tag{28}$$

Since $p_{Z|X}(z|x) = p_{X|Z}(x|z)$, we have $T_\rho f(z) = \mathbb{E}_x[f(x)|Z = z]$. And so, ignoring the $T_\rho f = 0$ case

$$\mathrm{sign}(T_\rho f(z)) = 1 \Leftrightarrow T_\rho f(z) > 0 \Leftrightarrow \Pr_x(f(x) = 1|Z = z) > 1/2 \Leftrightarrow g(z) = 1 \tag{29}$$

$\square$

For the interested reader, we provide a simple combinatorial proof for part of Proposition 1 of Weinberger & Shayevitz (2018).

**Lemma 5.** Fix a boolean function $f$ and generate $(X, Z)$ according to the bitflip scheme above. Then,

$$\Pr_{X|Z=z}(f(z) = f(X)) \geq \Pr_{X|Z=z}(f(z) \neq f(X)), \tag{30}$$

for all $z$ implies that $f \in \mathcal{F}_N^*(f)$.

*Proof.* We will show that the condition in Eq. 30 is sufficient for optimal prediction on noisy $Z$ by showing that no other boolean function can do better than $f$ when this condition holds. For any particular function $g$, define the set of inputs on which $g$ agrees with $f$ as

$$\mathcal{P}(f, g) := \{z \in \{0, 1\}^n : g(z) = f(z)\}, \tag{31}$$

We may rewrite the output of $g$ as

$$g(z) = \begin{cases} f(z), & z \in \mathcal{P}(f, g) \\ \neg f(z), & z \in \mathcal{P}(f, g)^c \end{cases} \tag{32}$$

Thus we find

$$\Pr_{X,Z}(g(Z) = f(X)) = \Pr_Z \Pr_{X|Z}(g(Z) = f(X)) \tag{33}$$

$$= \frac{1}{2^n} \sum_{z \in \{0,1\}^n} \Pr_{X|Z=z}(g(z) = f(X))$$

$$= \frac{1}{2^n} \sum_{z \in \mathcal{P}(f,g)} \Pr_{X|Z=z}(f(z) = f(X)) + \frac{1}{2^n} \sum_{z \in \mathcal{P}(f,g)^c} \Pr_{X|Z=z}(\neg f(z) = f(X))$$

$$= \frac{1}{2^n} \sum_{z \in \mathcal{P}(f,g)} \Pr_{X|Z=z}(f(z) = f(X)) + \frac{1}{2^n} \sum_{z \in \mathcal{P}(f,g)^c} \Pr_{X|Z=z}(f(z) \neq f(X))$$

$$\leq \frac{1}{2^n} \sum_{z \in \mathcal{P}(f,g)} \Pr_{X|Z=z}(f(z) = f(X)) + \frac{1}{2^n} \sum_{z \in \mathcal{P}(f,g)^c} \Pr_{X|Z=z}(f(z) = f(X))$$

$$= \Pr_{X,Z}(f(Z) = f(X)) \tag{34}$$

where the final inequality follows from Eq. 30, and we have used the fact that the marginal distribution of $Z$ is uniformly random whenever the distribution of $X$ is uniformly random. $\square$

Note that Eq. 30 for optimality at noisy prediction is equivalent to

$$\Pr_e(f(z) = f(z \oplus e)) \geq \frac{1}{2} \tag{35}$$

Consider the dictator function DICT, which outputs the value of the first bit. It immediately follows that for $f = $ DICT, $f_N^* = $ DICT, since

$$\Pr_e(\text{DICT}(z) = \text{DICT}(z \oplus e)) = 1 - \epsilon \geq \frac{1}{2}. \tag{36}$$

We will next show that Eq. 35 holds for PARITY as well.

**Proposition 6** (PARITY is optimal for predicting noisy PARITY)**.**

*Proof.* We will first show that for any noisy bitstring $z$, the parity of the corresponding noiseless bitstring is most likely to match the parity of $z$. Using linearity and $p < \frac{1}{2}$, we have

$$\Pr_{X|Z=z}(\text{PARITY}(Z) = \text{PARITY}(X)) := \Pr_E(\text{PARITY}(z) = \text{PARITY}(z \oplus E)) \tag{37}$$

$$= \Pr_E(\text{PARITY}(z) = \text{PARITY}(z) \oplus \text{PARITY}(E)) \tag{38}$$

$$= \Pr_E(\text{PARITY}(E) = 0) \tag{39}$$

$$= \Pr_E(\text{wt}(E) \text{ is even}) \tag{40}$$

$$= \frac{1}{2}(1 + (1 - 2p)^n) \tag{41}$$

The final line is never less than $\frac{1}{2}$ and therefore satisfies Eq. 35. By Lemma 5 the proof is complete.
$\square$

A combinatorial proof also exists showing that $f := \text{MAJ}_n$ satisfies $f = f_N^*$ for odd $n$. The proof is not particularly illuminating, so we have excluded it.

### A.4 PROOF OF PROPOSITION 2

Here we prove Proposition 2 and then discuss further the inequality

$$\text{I}[f] \underset{?}{\geq} \text{I}[f_N^*], \tag{42}$$

which holds with high probability for randomly sampled boolean functions (even for small $n$), but does not hold for all boolean functions.

Before proving an asymptotic relationship between the influence of $f_N^*$ and $f$, we will need statistical moments of random boolean functions. Recall that a Rademacher random variable takes values in $\{1, -1\}$ with equal probability. Each Fourier coefficient

$$\hat{f}(S) = \sum_{x \in \{-1,1\}^n} f(x)\chi_S(x) \tag{43}$$

is a sum of $2^n$ independent Rademacher variables, since each $f(x)$ is itself Rademacher. As a result, the coefficients are distributed according to $\hat{f}(S) \sim 2^{-n}(\text{Bin}(2^n, \frac{1}{2}) - 2^n)$ where Bin denotes the binomial distribution. Using standard results for the Binomial distribution yields

$$\mathbb{E}_f[\hat{f}(S)] = 0 \tag{44}$$

$$\mathbb{E}_f[\hat{f}(S)\hat{f}(T)] = \begin{cases} 2^{-n} & S = T \\ 0 & \text{else} \end{cases} \tag{45}$$

and so the Fourier coefficients of a random function are, on average, uncorrelated (they are not independent, as we require $\sum_{S \subseteq [n]} \hat{f}(S)^2 = 1$). We may now prove the following proposition, which leads directly to Proposition 2 in the main text.

**Proposition 7.** Let $f : \{1, -1\}^n \to \{1, -1\}$ be a uniformly random boolean function on $n$ bits, and define $f_N^* := \text{sign}(T_\rho f)$ for $\rho \in (0, 1)$. Then, for $r(\rho) = \frac{(1-\rho^2)}{(1+\rho^2)}$

$$\text{I}[f_N^*] = \frac{n}{\pi} \arccos(r(\rho)) \tag{46}$$

*Proof.* We will work with bitstrings taking values in $\{1, -1\}^n$. Define $k_\rho(z|x) := \Pr(Z = z|X = x)$ denote the probability for a noisy bitstring $z$ given a $\rho$-correlated input bitstring $x$, i.e. $k_\rho(z|x) = 2^{-n}(1 - \rho)^{\text{wt}(e)}(1 + \rho)^{n-\text{wt}(e)}$, and so $T_\rho f(x) = \sum_{z \in x} k_\rho(z|x)f(z)$. For uniformly random $f$, $f(z)$ is Rademacher, and we define a family of two-dimensional random variables $\{A_z\}$

$$A_z := \begin{pmatrix} k_\rho(z|x) \\ k_\rho(z|x^{\oplus i}) \end{pmatrix}. \tag{47}$$

For any particular $f$, we have $\mathbb{E}_z[A_z] = (T_\rho f(x), T_\rho f(x^{\oplus i}))^T$. By the Central Limit Theorem, we have that $\mathbb{E}_z[A_z]$ approaches a bivariate Gaussian distribution with covariance matrix

$$\Sigma := \mathbb{E}_z[A_z A_z^T] = \begin{pmatrix} \mathbb{E}_z[T_\rho f(x)^2] & \mathbb{E}_z[T_\rho f(x)T_\rho f(x^{\oplus i})] \\ \mathbb{E}_z[T_\rho f(x)T_\rho f(x^{\oplus i})] & \mathbb{E}_z[T_\rho f(x^{\oplus i})^2] \end{pmatrix}. \tag{48}$$

By symmetry, we only need to compute two terms in this matrix. Recall that

$$T_\rho f(x) = \sum_{S \subseteq [n]} \rho^{|S|} \hat{f}(S)\chi_S(x), \tag{49}$$

and so

$$\mathbb{E}_f[T_\rho f(x) T_\rho f(y)] = \mathbb{E}_f\left[\sum_{S,T\subseteq[n]} \rho^{|S|+|T|}\hat{f}(S)\hat{f}(T)\chi_S(x)\chi_T(y)\right] \tag{50}$$

$$= \sum_{S,T\subseteq[n]} \rho^{|S|+|T|}2^{-n}\mathbb{I}\{S=T\}\chi_S(x)\chi_T(y) \tag{51}$$

$$= 2^{-n}\sum_S \rho^{2|S|}\chi_S(x+y) \tag{52}$$

$$= 2^{-n}\prod_{i=1}^n (1+\rho^2 x_i y_i) \tag{53}$$

where we have used Eq. 45 in the second equality. The terms in the covariance matrix $\Sigma$ are then

$$\mathbb{E}_f[T_\rho f(x)^2] = \mathbb{E}_f[T_\rho f(x^{\oplus i})^2] = 2^{-n}(1+\rho^2)^n \tag{54}$$

$$\mathbb{E}_f[T_\rho f(x) T_\rho f(x^{\oplus i})] = 2^{-n}(1-\rho^2)(1+\rho^2)^{n-1}. \tag{55}$$

Thus, we may define a unit-variance variable $A'_z := A_z/\sqrt{2^{-n}(1+\rho^2)^n}$ which converges to a bivariate Gaussian distribution with covariance matrix

$$\Sigma' = \begin{pmatrix} 1 & \frac{1-\rho^2}{1+\rho^2} \\ \frac{1-\rho^2}{1+\rho^2} & 1 \end{pmatrix} \tag{56}$$

Then, $\Pr_x(\text{sign}(T_\rho f(x)) = \text{sign}(T_\rho f(x^{\oplus i})))$ is equal to the probability that both components of $A_z$ are positive. Clearly this probability is unaffected by renormalization, and so we may apply Sheppard's formula (e.g. O'Donnell (2014)): For standard Gaussian random variables $a_1, a_2$ with $\mathbb{E}[a_1 a_2] = r$,

$$\Pr(a_1 \leq 0, a_2 \leq 0) = \frac{1}{2}\left(1 - \frac{\arccos(r)}{\pi}\right). \tag{57}$$

Thus, we find that in the limit of large $n$, the average sensitivity of $f_N^*$ for a randomly sampled $f$ obeys

$$\mathbb{E}_f[\text{Inf}_i[f_N^*]] = 1 - \Pr_f\left(\text{sign}(T_\rho f(x)) = \text{sign}(T_\rho f(x^{\oplus i}))\right) \tag{58}$$

$$= 1 - 2\Pr_f(T_\rho f(x) \geq 0, T_\rho f(x^{\oplus i}) \geq 0) \tag{59}$$

$$\approx \frac{\arccos(r(\rho))}{\pi}. \tag{60}$$

Where we have invoked symmetry in the distribution of $f$ to remove the averaging over $x$ in the definition of $\text{Inf}_i$, and further invocation of symmetry gives $\text{I}[f] = n\,\text{Inf}_i[f]$. Substitution of $\rho = 1 - 2p$ recovers the result from the main text. $\square$

**Additional observations** We found that the inequality 42 holds for all Boolean functions of $n \leq 4$ bits, and a large number of randomly-sampled boolean functions for larger $n$. For $n \geq 5$, we tested this inequality with the following procedure: (1) sample a uniformly random element of $\{-1,1\}^{2^n}$, (2) generate $f$ from this vector as a hash table, (3) compute $T_\rho f$ in the Fourier domain, (4) compute $\text{I}[f_N^*]$ and $\text{I}[f]$ in the Fourier domain. We repeated this procedure for several million boolean functions for $n \in \{5,6,7,8\}$, for bitflip values in $\{0.01, 0.25, 0.49\}$. No randomly sampled function violated inequality 42.

Despite holding with high probability for random boolean functions, Eq. 42 does not hold for all boolean functions. A counterexample with $\text{I}[\text{sign}(T_\rho f)] > \text{I}[f]$ is given by the linear threshold function (LTF) $f(x) = \text{sign}(a_0 + \sum_{i=1}^n a_i x_i)$ with

$$(a_0, a_1, a_2, a_3, a_4, a_5, a_6) = (0.3, 0.1, 0.1, 0.2, .3, 0.4, 0.9) \tag{61}$$

and $\rho = 0.2$. Additional counterexamples may be found via grid search of LTFs over $\rho$ and $(a_i)$ for larger $n$. We are unaware of any characterization of all such functions violating Eq. 42.

A.5 DISCUSSION OF OTHER NOISE MODELS

We discuss how changes in the noise model might affect our findings. First, we argue that if the noise is non-independent, then the relationship between $I[f_N^*]$ and $I[f]$ can easily break down as the noise on distinct bits starts to become more correlated. Second, we argue that independent (but non-identical noise) is less likely to affect our findings.

We first generalize the Bayes-optimal predictor to handle non-iid noise. An arbitrary noise distribution is defined according to a conditional probability $q(z|x) := \Pr(Z = z|X = x)$ of a noisy bitstring $z$ given noiseless input $x$. If $z$ is generated by flipping bits of $x$ independently of the value of $x$, then we may write $q(z|x) := p_E(z - x)$. The generalization of the noise operator $T_\rho$ is then

$$(T_q f)(x) := E_{Z|X}[f(Z)|X = x] \tag{62}$$

$$= \sum_z q(z|x)f(z) \tag{63}$$

$$= (p_E * f)(x), \tag{64}$$

where $(f * g)$ denotes convolution. As in Lemma 4, one may verify that this generalized noise operator obeys $f_N^* := \text{sign}(T_q f)$. We may use Fourier theory to show that correlated noise can greatly increase the sensitivity of $T_q f$ over the sensitivity of $f$. Applying the convolution theorem to $T_q f(x) = (p_E * f)(x)$ gives O'Donnell (2014):

$$\widehat{T_q f}(S) = \hat{p}_E(S)\hat{f}(S) \tag{65}$$

The influence of $f$ is $I[f] = \sum_{S \subseteq 1,\dots,n} |S||\hat{f}(S)^2$, and so

$$I[T_q f] = \sum_{S \subseteq 1,\dots,n} |S|\hat{p}_E(S)^2 \hat{f}(S)^2 \tag{66}$$

A distribution $p_E$ with strong correlations between individual bit flip events corresponds to large values of $\hat{p}_E(S)$ for higher-order terms (larger $|S|$), in which case $I[T_q f]$ can become much larger than $I[f]$. While this argument is incomplete, the inequality $I[T_q f] > I[f]$ will often result in an inequality $I[\text{sign}(T_q f)] > I[f]$.

On the other hand, noise that is independent but non-identical will tend preserve the relationship between the sensitivity of $f$ and the sensitivity of $f_N^*$ demonstrated in Proposition 2. If the noise is uncorrelated, then $p_E$ is separable over bits, and so $\hat{p_E}(S) = \prod_{i \in S} \rho_i$ where $\rho_i$ is the correlation between $z_i$ and $x_i$. So $T_q f$ acts like an exponential filter in Fourier space, i.e.

$$\widehat{T_q f}(S) = \prod_{i \in S} \rho_i \hat{f}(S) \tag{67}$$

This is qualitatively similar behavior to the iid case where $\widehat{T_\rho}(S) = \rho^{|S|}$ results in $\mathbb{I}[T_\rho f] \leq I[f]$, suggesting that our observations about the relative sensitivity of $f$ versus $f_N^*$ (and specifically Proposition 2) likely extend to the case where noise is independent but non-identical across bits.

# APPENDIX B   EXPERIMENTAL METHODS

In this appendix, we describe in detail all the experimental methods used to obtain the results displayed in figures 1, 2, and 3.In what follows, we shall refer to the experiments performed to obtain the robustness results of section 3.1 as *experiment 1*, whereas *experiment 3* shall refer to all the methods used to show how one can trap transformers with simplicity bias (section 4).

## B.1   DATASET GENERATION

We generate synthetic datasets for our experiments. for each dataset, we specify the number of validation samples $n_{val}$, the number of training samples $n_{train}$, the number of bits in one data point $n_{bit}$, a bit flip rate $p$, a seed for reproducibility and a boolean function $f$. Given these inputs, we generate the following:

1. *Noiseless Data*: uniformly sample $n_{\text{train}} + n_{\text{val}}$ many bitstrings, each of length $n_{\text{bit}} - 1$, compute the final bit for each using $f$. Take the first $n_{\text{train}}$ as the training data and the rest as validation data.

2. *Noisy Data*: Take the noiseless data above, for each data point, apply a symmetric bit flip rate $p$ on each bit for the previous $(n_{\text{bit}} - 1)$ bits, and keep the last bit unchanged.

**Experiment 1** To obtain the results in Fig. 1, the function $f$ is taken as MAJ$(20, 5)$, MAJ$(40, 5)$, MAJ$(50, 3)$, and PARITY$(20, 4)$, respectively. For MAJ$(20, 5)$, MAJ$(40, 5)$, and MAJ$(50, 3)$, we chose $n_{\text{train}} = 2000$ and $n_{\text{val}} = 10000$, while for the function PARITY$(20, 4)$ we worked with $n_{\text{train}} = 5000$ and $n_{\text{val}} = 10000$.

**Experiment 2** To obtain the results in Fig. 2, we generated 3200 random $k$-juntas (boolean functions depending on only $k$ bits) for $k \in \{5, 6, 7\}$. All experiments used a total of $n = 10$ bits, with the subset of $k$ bits chosen randomly. We generated noisy input data using a bitflip rate uniformly sampled from

$$\{0, 0.05, 0.08, 0.10, 0.13, 0.16, 0.18, 0.2, 0.22, 0.24, 0.26, 0.28, 0.3\}, \tag{68}$$

At bitflip rates lower than $\approx 0.10$, we found that $f = f_N^*$ with high likelihood for our range of $k$ values. We generated training and validation sets with $n_{\text{train}} = 5000$, $n_{\text{val}} = 10000$ datapoints (smaller training datasets frequently resulted in underfitting).

**Experiment 3** For experiments shown in Fig. 3, we generated a "trap" function, i.e., a function $f$ for which the corresponding $f_N^*$ is such that $\text{err}_f(f_N^*) \approx \text{err}_f(f)$ and $I[f_N^*] \ll I[f]$. To construct this function, we proceeded as follows. First, we restricted our search space to boolean functions $f : \{0, 1\}^n \to \{0, 1\}$ satisfying $f(x) = f(\text{wt}(x))$ (weight-based function), since we only have $2^{n+1}$ such functions compared to the $2^{2^n}$ number of all possible boolean functions over $n$ bits. Let us denote the space of weight-based functions by $\mathcal{W}_n$. For each function in $\mathcal{W}_n$ (which can be completely characterized by an n-bit string $s$), we choose a bitflip rate $p \in (0, 1)$. We iterated over the values $n \in \{4, 5, 6, 7, 8\}$ and $p \in \{0.2, 0.22, 0.24, 0.26, 0.28, 0.30\}$. For each combination of $n$ and $p$, we initially computed $\text{err}_f(f_N^*)$, $\text{err}_f(f)$, $I[f]$, and $I[f_N^*]$. Since $\text{err}_f$ represents generalization error over an arbitrarily large dataset, for functions satisfying $\text{err}_f(f_N^*) \approx \text{err}_f(f)$, we sampled finite training and validation datasets to select a dataset such that the performance of a lookup table (which achieves optimal performance on validation data) was close to $\text{err}_f(f)$. This led to a specific dataset on $n = 8$, $p = 0.2$, where $f(i) := s_i$ with $s = 000110000$. We again use sparse functions depending on a subset $n$ of $n_{\text{bits}} = 14$ that depends only on the embedded string $s$. The individual models (SAN, RNN, and regularized SAN) were trained using $n_{\text{train}} = 10000$ and $n_{\text{val}} = 20000$. The resulting trap function that was used to generate the datasets used in experiment 3 will be denoted by $\mathcal{T}$.

## B.2 TRAINING AND HYPERPARAMETERS

Our experiments involve learning discrete functions, which can be sensitive to parameter initialization and hyperparameter choice. In order to control for the effects of these choices, we chose to perform many trials of each learning experiment (with different initializations) over a range of hyperparameters. In order to compare results across models and learning tasks, we used the following procedure to select hyperparameters for each experiment:

1. Select an initial range of hyperparameters

2. Perform several hundred trials of the noiseless learning experiment (to learn either $f$ or some randomly sampled $f$)

3. Prune the hyperparameter set to remove any ranges of hyperparameters for which models consistently failed to learn the target function

The result was a hyperparameter set for which a large number (at least 70%) of the corresponding set of models were capable of learning the noiseless function. This implies that failure to learn the function (in the noisy setting) is not a capacity problem, and instead is due to either bad initialization or limitations of the architecture. The exception for this procedure was learning PARITY$(20, 4)$ (Fig. 1d) which resulted in a bimodal distribution of models: Models were either able to learn parity, or completely failed early in the experiment. This is in line with results reported in Ref. Bhattamishra

et al. (2023b), which observed that $\text{PARITY}(20, 4)$ was near the limit of what is learnable by (small) transformers.

All experiments were performed on a CPU cluster consisting of either Intel Xeon Gold 6230 processors (20 cores, 2.1 GHz) and 768 GB of RAM each, or AMD EPYC 9754 processors (128 cores, 2.25 GHz) and 1.5 TB of RAM. On average, a single hyperparameter tuning experiment took approximately 5 hours across 300 cores in parallel. Total CPU estimates are

- $\sim 60,000$ CPU hours for Fig. 1 ($\sim 100,000$ CPU hours for Experiment 1 in total)
- $\sim 15000$ CPU hours for Fig. 2 ($\sim 30000$ CPU hours for Experiment 2 in total)
- $\sim 5000$ CPU hours for Fig. 3 ($\sim 10000$ CPU hours for Experiment 3 in total).

In each experiments, models were trained for up to 1000 epochs, with an early stopping criterion of 300 epochs conditioned on no improvement in validation accuracy. All reported metrics correspond to the model performance at the epoch with maximum validation accuracy. We do not tokenize any of these datasets, since both the underlying distribution and noise acting on our synthetic data depend on individual bits.

### B.3 MODEL ARCHITECTURES

**Transformer**: Our experiments employ an encoder-only transformer with a final linear layer to generate outputs. We initialize with the Xavier uniform Glorot & Bengio (2010) (following Ref. Bhattamishra et al. (2023b)) and ReLU activation for the encoder's FFN and absolute positional encoding. We train using cross-entropy loss. The initial hyperparameter search space for Experiments 1, 2, and 3 is given in Table 1, and refers to hyperparameters as introduced in Ref. Vaswani et al. (2017).

Table 1: Hyperparameter search space for Transformer model for experiments 1-3. The final hyperparameter sampling space across trials was tuned for each experiment separately (see above).

| Hyperparameter | Values |
|---|---|
| Learning Rate (`lr`) | $[1 \times 10^{-5}, 6.31 \times 10^{-2}]$ |
| Depth | 1, 2, 3, 4, 5, 6 |
| Model Dimension (`d_model`) | 16, 32, 64, 128 |
| Dropout Rate | 0.05, 0.10 |
| Number of Attention Heads | 2, 4, 8 |
| Feedforward Dimension (`d_ffn`) | 32, 64, 128 |

**LSTM**: We used LSTMs with Xavier uniform initialization, log-likelihood loss and softmax activation (i.e. cross-entropy loss). The initial hyperparameter search space for LSTMs is shown in Table 2.

Table 2: Hyperparameter search space for LSTM trials in Experiments 1 and 3.

| Hyperparameter | Values |
|---|---|
| Learning Rate (`lr`) | $[1 \times 10^{-4}, 2.78 \times 10^{-1}]$ |
| Embedding Size (`emb_size`) | 16, 32, 64, 128 |
| Hidden Size (`hidden_size`) | 16, 32, 64, 128 |
| Dropout Rate | 0.05, 0.10 |
| Depth | 1, 2, 3, 4, 5, 6 |

Our experiments are initialized with hyperparameters sampled from a distribution such that models achieved $\approx 70\%$ on the corresponding noiseless learning task. This introduces significant variance in model architectures and parameter counts. Tables 3-4 provide some typical parameter counts (and corresponding hyperparameters) for models initialized in this way in Experiment 1 for the $\text{PARITY}(20, 4)$ task.

Table 3: Parameter count and hyperparameter settings for three random transformers. Median parameter count for transformers was 32514 in the Experiment 1 PARITY$(20, 4)$ task.

| parameters | d_model | heads | d_ffn | depth |
|---|---|---|---|---|
| 42562 | 64 | 2 | 32 | 2 |
| 11218 | 16 | 4 | 32 | 5 |
| 8738 | 32 | 4 | 64 | 1 |

Table 4: Parameter count and hyperparameters for three random LSTMs from the Experiment 1 PARITY$(20, 4)$ task. Median parameter count: 46530.

| parameters | emb_size | hidden_size | depth |
|---|---|---|---|
| 54978 | 64 | 32 | 6 |
| 11938 | 64 | 16 | 4 |
| 29634 | 64 | 32 | 3 |

### B.3.1 EXPERIMENT 1

Experiment 1 involved training transformers and LSTMs to learn MAJ and PARITY from noisy features. For each of $f \in \{$MAJ$(20, 5),$ MAJ$(40, 5),$ MAJ$(50, 3),$ MAJ$(30, 4),$ PARITY$(20, 4)\}$, we construct datasets $(z, f(x))$ with a range of bitflip error probabilities selected from $[0, 0.49)$. For each dataset, we train 300 randomly initialized LSTMs and Transformers (each training run is one trial) to minimize their corresponding loss function, with early stopping and model selection based on maximum validation accuracy.

Fig. 4 shows distributions of validation accuracies for all experiments involving MAJ functions. For all majority functions, the transformer validation accuracy concentrates near optimal for each bitflip rate. By contrast, from Fig 3g the noiseless performance of transformers on MAJ$(30, 4)$ degrades faster than MAJ with odd $n$. We also find that the distribution of LSTM validation accuracies for MAJ$(30, 4)$ tends to be more bimodal than for MAJ problems with odd $n$.

For learning PARITY$(20, 4)$, Fig. 5 shows a clear bimodal distribution in the final validation accuracy across a range of bitflip rates. Other than the noiseless scenario, LSTMs trained in our hyperparameter set were practically incapable of noise-robust learning.

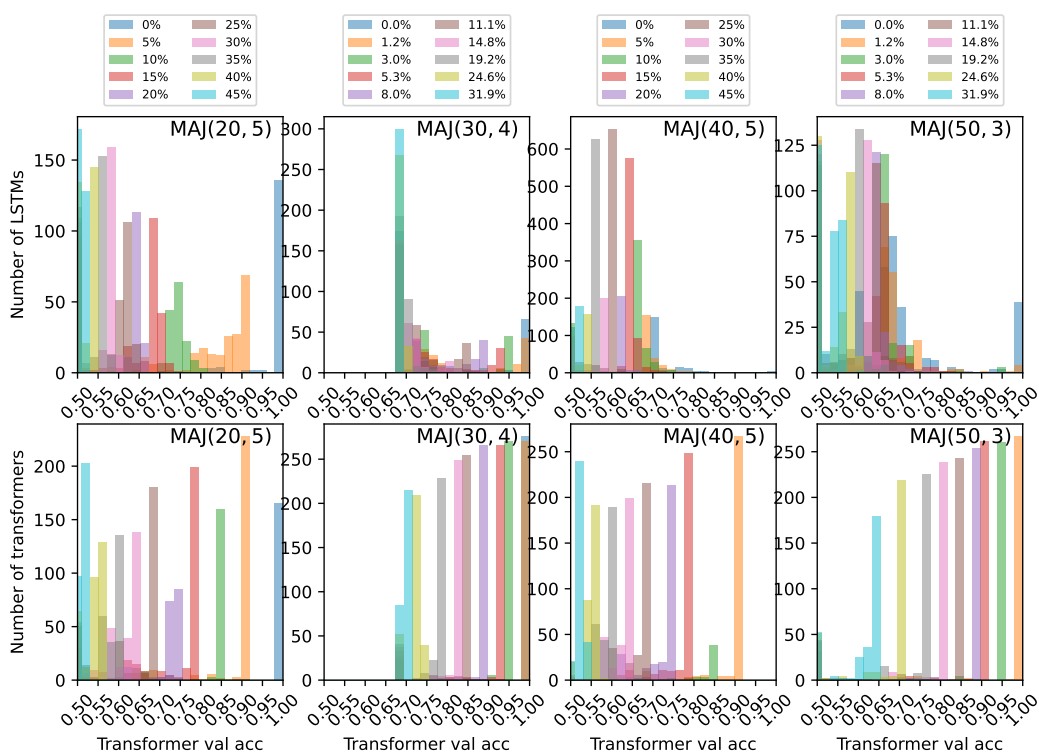

Figure 4: Validation accuracy for LSTMs (top row) and transformers (bottom row) for each sparse MAJ noise-robust learning task, colored according to bitflip rate. Each experiment with a particular error rate consists of 300 independent training runs with random initialization (maximum validation accuracy over each training trial is shown).

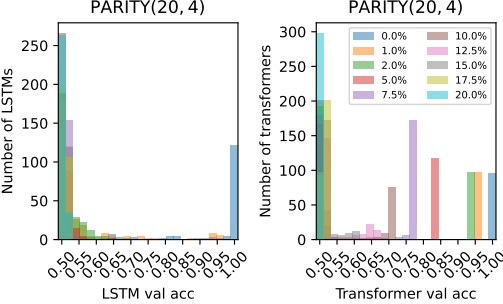

Figure 5: Validation accuracy for LSTMs (left) and transformers (right) for the PARITY(20, 4) noise-robust learning task (other details same as Fig. 4).

### B.3.2 EXPERIMENT 3

Figure 6 represents the collection of experiments that were used to show that transformers can be trapped due to their simplicity bias (refer to section 4). Fig. 6 compares the relationship between sensitivity and accuracy for different architectures for both noisy and noiseless validation datasets generated with the function $\mathcal{T}$. Each plot shows training trajectories of individual models (colored lines) that end at the point where the model achieves maximum accuracy on the respective validation dataset (circular markers). In the presence of noise (Figs. 6(a), 6(c), and 6(e)), one sees that all the models cluster in a relatively narrow accuracy band around $(0.5, 0.6)$. As pointed out in Section 4, the overall behavior of the SANs is to learn a function that optimizes the validation accuracy (optimal validation lookup table represented by ✕). Nonetheless, due to the unavoidable randomness in machine learning models, combined with the transformer simplicity bias, Fig. 6(a) shows that some individual models can get significantly closer to the true function (represented by ★). This is certainly not the case with LSTMs (Fig. 7), as all the individual models seem to cluster around the optimal training accuracy (lookup table for training set denoted by ✕). In Fig. 7(c), we see that the presence of simplicity penalty helps to bring the vast majority of the individual models significantly closer to the true function.

On the other hand, when the models are evaluated on the noiseless (original) dataset (Figs. 6(b), 6(d), and 6(f)), the architectures SAN and LSTM fail to learn the original function. Nonetheless, SANs perform better than LSTMs and are generally more suitable for this problem. As before, the addition of a simplicity penalty improves the learning traces of the SANs.

The pattern no longer holds for the function MAJ$(30, 4)$ (Fig. 7). Here, in both the noisy and noiseless validation cases, there seems to be no overall pattern followed by the individual learning traces. The overall behavior of the experiments indicates that LSTMs and SANs approach optimal validation accuracy, with no clear differences in their sensitivity. Adding a simplicity penalty to SANs has minimal effect on their performance. Across experiments, no model comes close to learning the true function.

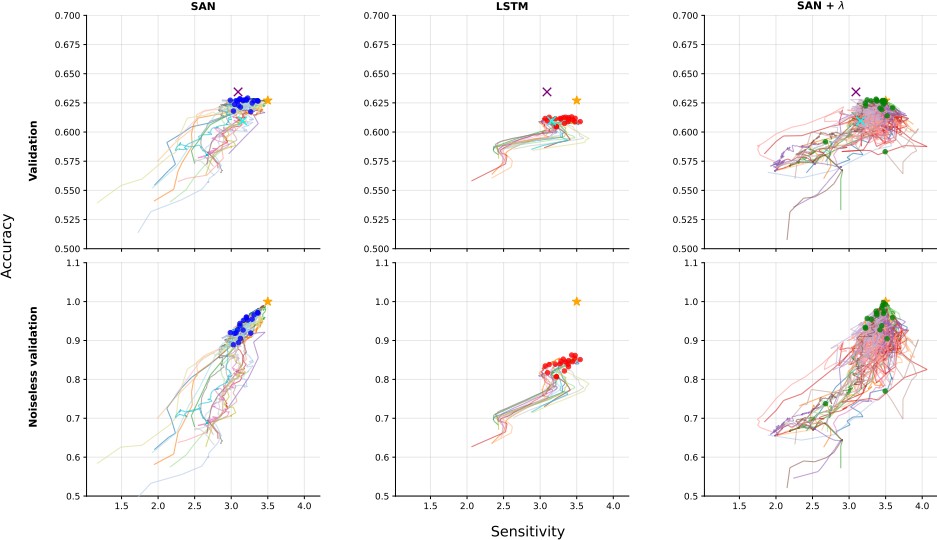

Figure 6: Validation (top row) and noiseless-validation (bottom row) accuracies for each architecture (rows: SAN, RNN, SAN+$\lambda$) trained on the dataset generated by the trap function $\mathcal{T}$ as a function of the sensitivity. Each colored curve is one of 19 independent runs, shown up to the epoch of its peak validation accuracy (blue, red, and green circles). The ✕ represents a lookup table for the validation dataset; the ✕ represents a lookup table on the training dataset; the ★ represents the true (noiseless) function.

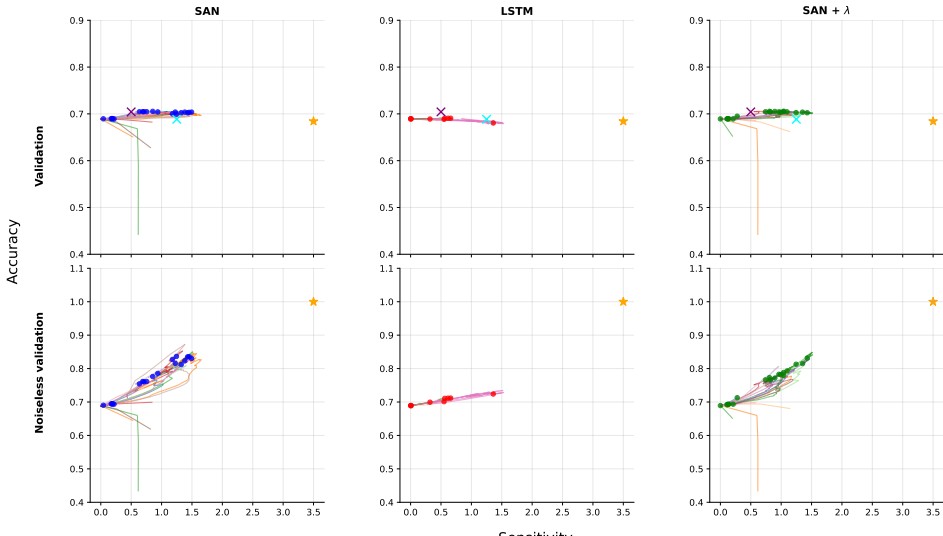

Figure 7: Validation (top row) and noiseless-validation (bottom row) accuracies for each architecture (rows: SAN, RNN, SAN+$\lambda$) trained on datasets generated with MAJ$(30, 4)$ as a function of the sensitivity. Each colored curve is one of 19 independent runs, shown up to the epoch of its peak validation accuracy (blue, red, and green circles). The ✕, ✕, ★ retain their meaning from Fig. 6.

## STATEMENT ON LLM USAGE

Large language models were used to assist in literature review, code development, theory development, and analysis for this project. None of the text or equations in this manuscript were generated by an LLM.

