# OpenReview forum: "Trapped by simplicity: When Transformers fail to learn from noisy features"
_ICLR.cc/2026/Conference — ICLR 2026 Poster_

### Official Review · Reviewer_NNLa · 2025-10-29

**Soundness:** 3
**Presentation:** 3
**Contribution:** 2
**Rating:** 6
**Confidence:** 3

**Summary:**

This paper studies the capability of transformers on noise-robust learning. Transformers are compared to LSTMs for this task. Main contributions of this paper include the following.
1. For sparse parity and majority functions at high rates of feature noise, transformers perform significantly better than LSTMs, and the latter fails even for low levels of feature noise.
2. They observed that transformers do not have satisfactory performance for the task of random k-juntas.
3. They proposed an explanation related to sensitivity of optimal solution and of target function.
4. Solution: implementing penalty for high-sensitivity solutions

**Strengths:**

1. The problem formulation is novel by adding a perspective on robustness. Noise-robust learning is valuable but yet relatively under explored.

2. The paper has sound theoretical analysis and the theoretical results provide significant insights. It deploys various mathematical tools efficiently, including Boolean analysis, information theory, and learning theory.

3. Experiments, although relatively small-scaled, has a clear target on the conjecture and provides valuable support

4. The paper is in general well-written and the logic flows smoothly

**Weaknesses:**

My main concern is on the applicability and scope of this study. The investigated problems (parity and junta) are binary-input problems with rigid mathematical structures. This fact provides simplicity for analysis, but at the same time they are restricted because real-world data and noises are much more complicated.

**Questions:**

My question is highly relevant with what I wrote in the weaknesses section: Does the theoretical insights obtained from your study have any implications on the robustness of transformers for more complicated tasks

---

> ### Author Response · Authors · 2025-11-21
>
> Thank you for taking the time to review our submission!
>
> > “Does the theoretical insights from this study have any implications on the robustness of transformers for more complicated tasks”
>
> At a high level, we expect the behavior we study here to be an issue in other domains where transformers are used. The phenomenon we predict via theory (and validate in experiments) is: (i) the optimal function for learning with feature noise $f_N^*$ tends to be simpler than the true boolean function $f$, (ii) transformers prefer simplicity, therefore (iii) transformers will not succeed at learning f from noisy features. We provide empirical evidence that step (i) holds for many boolean functions, and it seems likely that (i) also holds for at least some tasks on discrete domains (like text prediction), given the well-known smoothing/regularization effect of training with feature noise (Bishop, 1995). More evidence continues to appear for part (ii) for all sorts of simplicity measures and domains where transformers are applied. We hope that our relatively rigorous characterization of this phenomenon in an (admittedly) simple setup will motivate efforts towards studying more complex systems.
>
> For example, in audio or video processing with “input jitter” (essentially feature noise), our analysis suggests that an analogous effect should hold: The transformer should be likely to get "trapped" in a simple solution for the learning problem. The issue is, since video processing is so complex, we may not even be aware how to detect the transformer becoming trapped in a simple solution!
>
> From another angle, our results are immediately relevant to tasks where transformers are used to process binary data directly. For example, consider classical and quantum error correction. Decoding an error correcting code is essentially the following: A a 1-bit message b is encoded into a length-n bitstring x (“codeword”) such that f(x) = b. Then, noise acts on this bitstring x to generate a noisy codeword z. The goal of the decoder is to learn a boolean function g such that g(z) = f(x) with high probability over x,z, i.e. exactly our setting.
>
> Hopefully our submission will inspire investigation into the simplicity/sensitivity of transformers trained on noisy text features, so that we can better understand the effects of text corruption/semantic sloppiness upstream of the LLMs that are so prevalent now. There are plausible ways of running controlled experiments on this effect, for example studying the effects of corrupted text in a small NLP corpus like TinyStories on how LLMs learn in the presence of noisy features, but it will also require some progress on the theory side to approach this problem in a productive way.

---

### Official Review · Reviewer_RgtW · 2025-11-01

**Soundness:** 3
**Presentation:** 3
**Contribution:** 3
**Rating:** 6
**Confidence:** 2

**Summary:**

The paper studies noise-robust learning: whether models trained only on inputs corrupted by feature noise can still learn the underlying noiseless target function. Using Boolean tasks, the authors:

* Formalize the Bayes-optimal predictor under iid bit-flip noise as $f_N^*(x)=\mathrm{sign}(T_{1-2p} f(x))$, where $T_\rho$ is the standard noise operator. This links the problem to noisy-channel coding and bounds performance via conditional entropy.
* Show empirically that transformers (encoder-only SANs) often succeed at noise-robust learning for sparse parity and odd-length sparse majority functions, while LSTMs generally fail even at modest noise rates. (Figure 1)
* Demonstrate that transformers typically fail on random k‑juntas, even when they achieve near‑optimal accuracy on the noisy validation distribution; failure correlates with the gap between the sensitivity of $f$ and that of the optimal noisy predictor $f_N^*$. (Figure 2)
* Argue the failure mechanism combines a simplicity bias in transformers toward low‑sensitivity functions and the empirical observation that $f_N^*$ tends to have lower sensitivity than $f$. (Conjecture 1)
* Construct a controlled "trap" function where $f$ and $f\_N^\*$ have similar optimal noisy accuracy but very different sensitivities; transformers converge to the simpler $f\_N^\*$. Adding a differentiable sensitivity penalty helps "escape" in a narrow range of penalty weights, but when $f\_N^\*$ is much better than $f\_N^\*$, the penalty does not help.

**Strengths:**

* Casting training-on-noisy-features as a noisy-channel problem, with $f_N^*$ characterized by the noise operator and performance tied to $H(Y|Z)$, gives a precise target for comparison.
* Extensive hyperparameter sweeps and repeated trials (300 per condition) improve the reliability of the conclusions.
* The use of total influence $I[f]$ to quantify simplicity connects to prior theory and cleanly explains why models can perform well on noisy validation yet fail on noiseless evaluation.
* The “trap function” isolates simplicity bias from other confounders: with nearly equal noisy optimal errors for $f$ and $f_N^*$, any preference for $f_N^*$ reveals the inductive bias. Figure 8 show SANs drifting toward the trap unless regularized.
* The discussion connects Boolean results to real LLM settings where training data are noisy/stochastic but downstream tasks are noise‑sensitive (e.g., arithmetic), cautioning that minimizing conditional entropy on noisy features might impede learning fine‑grained rules.

**Weaknesses:**

* Narrow noise model and data distribution. All inputs are uniformly random bitstrings with iid symmetric bit‑flip noise and memoryless corruption. Real text has structured distributions and correlated, non‑binary errors (insertions, deletions, paraphrases). The paper acknowledges this but leaves generality uncertain.
* Task scope. Results hinge on Boolean functions; while parity/majority and k‑juntas are classic, evidence that the same mechanisms dominate in natural language or code remains indirect. No experiments on tokenized sequences, algorithmic datasets, or synthetic “text‑like” corruptions are provided.
* The sensitivity penalty that helps in the trap requires a narrow range of $\lambda$ and does not generalize when $f_N^*$ dominates $f$. This makes it difficult to apply similar ideas in practice
* Conjecture 1—$I[f_N^*] \le I[f]$—is well‑motivated and tested on small n and random samples, but remains unproven. Many conclusions rely on it conceptually; a counterexample would undercut the narrative.

**Questions:**

1. How sensitive are the main findings to non‑iid or structured noise (e.g., burst errors, deletions/insertions, or noise correlated with specific positions)? Could the channel view extend to these cases and does the simplicity gap persist?
2. What happens when inputs are non‑uniform (e.g., biased bit marginals or low‑entropy substructures), closer to natural text statistics? Does the prevalence of self‑predicting functions change under such distributions?
3. Can the authors provide explicit parameter counts to the models? It is a bit troublesome to compute them from the dimensions provided in the appendix. Does model size/capacity have any effect on the biases observed?
4. Is there a practical diagnostic to detect, during training, when a model is converging to $f_N^*$ rather than $f$? The training‑trace figures suggest this might be possible.

---

> ### Author Response · Authors · 2025-11-21
>
> Thanks so much for taking the time to review our submission! Below are responses to questions (part 1)
> ### 1. **non-iid noise**
>
> Good question! Basically, if the noise is independent but non-identical, this shouldn't affect the relationship between the sensitivity of $f$ and the sensitivity of $f_N^\*$ too much. But if the noise is non-independent, then the relationship between $I[f_N^\*]$ and $I[f]$ can break down as the noise on distinct bits starts to become more correlated.
>
> To see this, we first generalize the Bayes-optimal predictor to non-iid noise. An arbitrary noise distribution is defined according to a conditional probability $q(z|x)$ of producing a noisy bitstring $z$ given noiseless feature $x$. Assume that z is generated by bitflips of x, so that $q(z|x) = p_E(e)$ is independent of the input (this is not the most general case). The generalization of the noise operator $T_\rho$ is  $(T_q f)(z) := E_{z|x} [f(x)| z]$, and an identical technique to Appendix A.3 shows that $f_N^\*:= sign(T_q f)$.
>
> To see that correlated noise can greatly increase the sensitivity of $T_q f$ over the sensitivity of $f$, we use Fourier theory. The convolution theorem (O’Donnell, 2014) gives:
>
> $T_qf(x) = (p_E \* f)(x) \Rightarrow \widehat{T_q f}(S) = \hat{p}_E(S) \hat{f}(S)$
>
> The influence of $f$ is $I[f] = \sum_{S \subseteq {1,\dots,n}} |S| \hat{f}(S)^2$ , so
>
> $I[T_q f] = \sum_{S \subseteq {1,\dots, n}} |S| \hat{p}_E(S)^2 \hat{f}(S)^2$
>
> If $p_E$ is highly correlated - e.g. there is large probabilities for 3, 4, 5 bitflips to occur together - then $\hat{p}_E(S)$ gets larger for higher-order terms (larger $|S|$), so $I[T_q f]$ can become much larger than $I[f]$. This is only a sketch since we actually care about $I[sign(T_q f) ]$ and not $I[T_q f]$, but a lot of the sensitivity of $T_q f$ will persist through taking the sign function.
>
> On the other hand, if the noise is independent but not identical, then intuitively we will still usually expect to see $I[f_N^*]\leq I[f]$ and our analysis still holds. If the noise is uncorrelated, then $p_E$ is separable over bits, and so $\hat{p_E}(S) = \prod_{i \in S} \rho_i$ where $\rho_i$ is the correlation between $z_i$ and $x_i$. So $T_q f$ acts like an exponential filter in Fourier space, i.e.
>
> $\widehat{T_q f}(S) = \prod_{i \in S} \rho_i \hat{f}(S)$
>
> This is similar behavior to the iid case where $\widehat{T_\rho}(S) = \rho^|S|$, so our findings will largely hold. We will add some discussion of more general noise behaviors to the submission when we reupload.
>
> ### 2. **non-uniform inputs**
>
> The theory is trickier and so we don’t have as much to say there. Our guess is that if inputs are non-uniformly random – e.g. the first bit is biased towards “0” in the training data – then the model would not learn a strong dependence on that bit and would default towards having low sensitivity to that bit (recall that the default behavior of transformers with untrained parameters is low sensitivity, e.g. Bhattamishra et. al. 2023 and Vasudeva et. al. 2025).
>
> We tried some experiments for noise-robust learning on bitstrings from distributions that were more similar to text. We did so by building strings recursively, e.g. by defining a rule $h: {0,1}^4 \rightarrow {0,1}$ and then starting from a seed string like 1011, one can recursively build a length N string $(1,0,1,1, h(1,0,1,1), h(0,1,1,h(1011)), …)$. We ran the same experiments as Fig 1, i.e. train to minimize noisy validation error, then evaluate noiseless test accuracy. The results looked different from Fig. 1, with noiseless test accuracy falling off at a similar pace as noisy validation accuracy. But we didn’t end up including these results because:
>
> a. The recursive rule was contrived (most recursive rules result in short cycles like 1010101…, so we had to design a particular rule that had long cycle length)
>
> b. Computing the optimal prediction rule was intractable (the closed form for the boolean function induced on $N$ bits has exp(N) complexity in general).

---

> > ### Author Response · Authors · 2025-11-21
> >
> > (part 2)
> >
> > ### 3. **parameter counts**
> >
> > See the table below. We will update the submission with these additional details. Because of our hyperparameter sampling scheme total parameter count varies across different problems. These are examples of median parameter counts (and corresponding hypers) for models in the (20, 4) parity experiments:
> >
> > ```
> > SAN (Transformer): Median Param Count (32514)
> > parameter count |         dim model |               heads |           dim ffn |         depth
> >        42562 |                64.0 |                  2.0 |             32.0 |            2
> >        11218 |                16.0 |                  4.0 |             32.0 |            5
> >         8738 |                32.0 |                  4.0 |             64.0 |            1
> > ```
> > ```
> > RNN (LSTM): Median Param Count (46530)
> > param count |         emb size |       hidden size |         depth |
> >        54978   |         64.0 |            32.0 |                6 |
> >        11938   |         64.0 |            16.0 |                4 |
> >        29634   |         64.0 |            32.0 |                3 |
> > ```
> >
> > ### 4. **Tracking convergence**
> >
> > Yes, this should be possible. One way to detect whether the model is converging towards $f_N^*$ versus $f$ is cosine similarity $\langle f, \hat{f}\rangle$ (where $\hat{f}$ is the function the transformer learns). In Fig 3b and 3e this is implicit since noiseless test accuracy is just $(\langle f, \hat{f}\rangle + 1)/2$. It would be possible to make a plot of $\langle f, \hat{f}\rangle$ versus $\langle f_N^\*, \hat{f}\rangle$, though it wouldn’t contain any information about the sensitivity of of $\hat{f}$. Of course, this is not something one can do in practice to mitigate the effects we studied here.
> >
> >
> > ### Other comments on weaknesses
> >
> >  - **Task scope** - Yes, we only study next-bit prediction. Since the theory for this setting is specific to Boolean functions, we felt that including experiments on natural text might be out of scope (and also require a lot of additional setup to
> > Sensitivity penalty: It is true that this technique will not generalize when $f_N^*$ dominates $f$. We believe that this emphasizes that the limitation for transformers from learning from noisy data is intrinsic to their architecture: The simplicity bias is simply too strong to counteract with loss-based regularization. It will require an architecture with more specific biases to learn any particular boolean function from noisy features!
> >  - **Issue with Conjecture 1**: Indeed, we recently found a counterexample to Conjecture 1 a few weeks after making this submission. We believe that this does not affect the narrative of our work too much, since it remains true that the inequality in Conjecture 1 is only rarely violated, e.g. it holds with very high probability for randomly generated k-juntas, and holds for "common" boolean functions that one can think of. A sketch of how to prove this: a uniformly random Boolean function $f$ will have sensitivity close to $n/2$ and Fourier spectrum $\hat{f}(S) \approx 2^{-n}$ for all $S$. Using similar arguments as discussed above, this means that higher-order Fourier coefficients of $T_\rho f$, meaning that $sign(T_\rho f)$ shouldn't be close to $n/2$ sensitivity. We are revising the submission to discuss this refined observation, provide better statistics on how often this inequality is violated, and discuss these counterexamples in more detail.

---

> > > ### Comment · Reviewer_RgtW · 2025-11-26
> > >
> > > I thank the authors for the explanations and additional details. I will maintain my score.

---

### Official Review · Reviewer_TMxh · 2025-11-01

**Soundness:** 3
**Presentation:** 2
**Contribution:** 3
**Rating:** 6
**Confidence:** 3

**Summary:**

This work analyzes the ability of transformers and LSTMs in learning boolean functions, in the feature noise setting (with no label noise). The motivation for this lies in the prevalence of noise in LLM training data, and boolean functions serve as a sandbox for understanding this phenomenon. Through extensive simulations, it is shown that for the majority function and certain sparse parity functions, transformers succeed with learning the optimal function while LSTMs fail. On the other hand, for most other random boolean functions, both of these architectures fail to learn in the feature noise setting, although for the case of transformers, this can be mitigated with a modified regularized loss function.

**Strengths:**

- The motivation for the theoretical setup is clear (e.g. inspiration from modern day LLM training).
- Extensive experiments are provided for demonstrating when transformers can and cannot learn boolean functions (e.g. which $k$ in a $k$-sparse parity in which learning with feature noise is possible, as well as majority functions, and other $k$-juntas).
- The analysis of the sensitivity of the optimal predictor under feature noise versus the teacher is an interesting perspective, and a conjecture regarding this is proposed, backed by simulations.

**Weaknesses:**

- The analysis for parity and majority seem quite straightforward, and it would be interesting if there was more theoretical analysis on progress towards the conjecture.
- For the LSTM model, it would be useful to have some theoretical analysis of this setting for learning boolean functions too.
- Perhaps an example of a realistic setting of feature noise in training transformers would be useful, and I believe this would better motivate this paper.

**Questions:**

- What are some of the main bottlenecks preventing the authors from making the conjecture rigorous?

---

> ### Author Response · Authors · 2025-11-21
>
> Thanks for the kind review!
>
> Indeed we have made some (unexpected) progress on the conjecture - several weeks after making this submission, we discovered a counterexample while trying to prove the conjecture. And so, Eq (7) $I[f_N^*] \leq I[f]$ cannot be true in general. But it seems to be _usually_ true, as we never found a counterexample during many millions of test cases during our project, nor any $k$-junta with $k\leq 4$, nor majority or parity. So, we are updating the submission to replace the conjecture with an empirical observation that, with high probability with respect to uniformly random k-juntas, the inequality $I[f_N^\*] \leq I[f]$ holds.
>
> We are working on making this statement rigorous. Here is a sketch of why the inequality should be true on average: A typical, random boolean function $f$ has a flat Fourier spectrum and therefore $I[f]\approx n/2$. The effect of noise $T_\rho$ is to suppress higher frequencies that contribute to this sensitivity, so the remaining function $T_\rho f$ is likely to be low sensitivity. Finishing this argument involves taking the sign function, and its still a work in progress.
>
> While the existence of counterexamples to this inequality makes our argument slightly weaker, the fact that such counterexamples are so rare means that the inequality may still hold more often than not for next-bit prediction tasks. This development also raises questions of independent theoretical interest, like what are the properties of boolean functions that violate this inequality, and does this relationship become more likely or less likely once we move to more realistic learning tasks involving natural text?
>
> **More details**: The family of counterexamples that we found were Linear Threshold Functions (LTF), which have the form $f(x) = sign(a_0 + a_1 x_1 + \cdots a_n x_n)$, for some fixed choice of $(a_0, \dots, a_n)$. We had initially tried to prove the counterexample for LTFs, but when that didn’t work we performed a grid search over many $\mathbf{a}$ vectors for different noise rates $\rho$, $n$, and eventually found a violation at $n=6, \rho=0.2$. Repeating this process yields additional counterexamples for $n=7,8,9$, and so presumably examples exist for all $n$ sufficiently large.
>
> Many LTFs violating Eq. (7) follow this pattern: One $a_i$ is relatively large, and the remaining coefficients $a_j$ for $j\neq i$ are smaller but non-negligible. As a result, the influence of bit i on $f_N^\*$ is smaller than for $f$, but the influence of bit $j\neq i$ gets slightly larger. And so when n is large, these small increases in bitwise influence contribute more to $I[f_N^*]$ than the decrease in influence of bit i.
>
>
> ### Other weaknesses
>
>  - **A realistic setting for feature noise in training transformers** We will update the submission to motivate our line of work more directly. Two settings directly involving training transformers to learn boolean functions with feature noise  are:
> 1. Classical and quantum error correction: Decoding an error correcting code is, in the simplest possible setting, the following: A a 1-bit message b is encoded into a length-n bitstring x (“codeword”) such that f(x) = b. Then, noise acts on this bitstring x to generate a noisy codeword z. The goal of the decoder is to learn a boolean function g such that g(z) = f(x) with high probability over x,z, which is exactly our setting.
> 2. Learning parity with noise: this is equivalent learning sparse-parity with feature noise, for some choice of feature noise. This problem has been independently interesting in the learning theory community for a while, and its (un)solvability has implications for how well transformers can break cryptographic protocols.
>  - **Theory for the LSTM**: most of the theory we showed was for boolean functions in general, and should hold for the output of either an LSTM or transformer. And so we weren't sure which theoretical analysis you had in mind here - if there was a specific analysis you were interested, could you please clarify?

---

### Meta-Review · Area_Chair_T3sK · 2026-01-02

**Summary:**

Each reviewer agreed that the contributions of the paper were interesting, both the theoretical parts and the supporting experiments.  The main concern of each reviewer was the general 'applicability' of the authors' results to more realistic settings, or for more unusual noise distributions.  I think that addressing these concerns would improve the paper, but believe the contributions are sufficient for acceptance, which I recommend.

**Reviewer Concerns:**

TMxH was concerned about the lack of a proof for Conjecture 1 in the original submission, and the authors followed up with a note that they had found a counterexample and thus revised the paper to address that.

I think the concerns about the "realism" of the settings were reasonably addressed.

The authors had some reasonable sketches for how things may work in more complex noise models which would help address the concern of reviewer RgtW.

**Reviewer Scores:**

I believe each reviewer would have maintained their scores.  TMxh appreciated the authors' LTF counterexample; RgtW explicitly mentioned they would; while NNLa mainly had concerns with applicability, which can't reasonably be resolved in a rebuttal.

---

### Decision · Program_Chairs · 2026-01-26

Accept (Poster)